# Variants in *SART3* cause a spliceosomopathy characterised by failure of testis development and neuronal defects

Katie L. Ayers [1,2] ✉, Stefanie Eggers[3], Ben N. Rollo[4], Katherine R. Smith[5], Nadia M. Davidson[5,6,7], Nicole A. Siddall [8], Liang Zhao [9], Josephine Bowles [9,10], Karin Weiss[11], Ginevra Zanni[12], Lydie Burglen[13,14], Shay Ben-Shachar[15], Jenny Rosensaft[16], Annick Raas-Rothschild[17,18], Anne Jørgensen[19], Ralf B. Schittenhelm [20], Cheng Huang [20], Gorjana Robevska[1], Jocelyn van den Bergen[1], Franca Casagranda[8], Justyna Cyza[1], Svenja Pachernegg[1,2], David K. Wright[4], Melanie Bahlo[5,36], Alicia Oshlack [21,22], Terrence J. O'Brien[4,23], Patrick Kwan[4,23], Peter Koopman[9], Gary R. Hime [8], Nadine Girard [24], Chen Hoffmann[25], Yuval Shilon[26], Amnon Zung[27,28], Enrico Bertini[12], Mathieu Milh [24], Bochra Ben Rhouma[29,30], Neila Belguith[30,31], Anu Bashamboo[32], Kenneth McElreavey[32], Ehud Banne[16,33], Naomi Weintrob[18,34], Bruria BenZeev[35] & Andrew H. Sinclair [1,2]

Squamous cell carcinoma antigen recognized by T cells 3 (*SART3*) is an RNA-binding protein with numerous biological functions including recycling small nuclear RNAs to the spliceosome. Here, we identify recessive variants in *SART3* in nine individuals presenting with intellectual disability, global developmental delay and a subset of brain anomalies, together with gonadal dysgenesis in 46,XY individuals. Knockdown of the *Drosophila* orthologue of *SART3* reveals a conserved role in testicular and neuronal development. Human induced pluripotent stem cells carrying patient variants in *SART3* show disruption to multiple signalling pathways, upregulation of spliceosome components and demonstrate aberrant gonadal and neuronal differentiation in vitro. Collectively, these findings suggest that bi-allelic *SART3* variants underlie a spliceosomopathy which we tentatively propose be termed INDYGON syndrome (**In**tellectual disability, **N**eurodevelopmental defects and **D**evelopmental delay with 46,X**Y GON**adal dysgenesis). Our findings will enable additional diagnoses and improved outcomes for individuals born with this condition.

Differences of sex development (DSD) are a heterogeneous group of congenital conditions that affect the reproductive system[1]. 46,XY gonadal dysgenesis is a rare DSD in genetic males characterised by partial or complete disruption to testis development and external genitalia ranging from underdeveloped male to typical female (MIM 400044, 1·9 cases per 100,000 live births[2,3]). Causative genetic variants are reported in only 35–45% of 46,XY DSD patients[4,5] and a

handful of genes are diagnostic for 46,XY gonadal dysgenesis with extragonadal comorbidities (ranging from neurodevelopmental defects to heart, kidney and skeletal anomalies) e.g. *ARX (MIM 300382), ATRX (MIM 300032) DHH (MIM 605423), SOX9 (MIM 608160)* and *WT1 (*MIM 607102)[6–10]. An early molecular diagnosis is clinically beneficial for individuals born with a DSD as it may inform patient clinical management in relation to adrenal and gonadal function, gender

development and gonadal cancer risk[11,12]. Importantly, since DSDs are frequently detected at birth, often before the emergence of comorbidities, a DSD presentation may trigger early genetic investigations that could result in diagnosis of a broader congenital syndrome. Early diagnosis permits timely intervention and access to disability, educational or social services. However, accurate genetic diagnosis of 46,XY gonadal dysgenesis and associated syndromes is currently hindered by significant gaps in our understanding of the genes and pathways involved.

The spliceosome is responsible for recognising and promoting precise splicing of the non-coding introns in precursor messenger RNA (mRNA). Pathogenic variants in spliceosome components cause conditions termed spliceosomopathies, in which brain defects, craniofacial anomalies and intellectual disability are common features[13]. Recently, pathogenic variants in the *TOE1* gene have been shown to cause defects in spliceosomal snRNA processing, leading to Pontocerebellar Hypoplasia Type 7 (OMIM 614969), a syndrome characterised by neurodegeneration and DSD including abnormalities of the gonads or external genitalia[14]. The relationship between disruption of the spliceosome and abnormal gonadal development is only now emerging and the underlying pathogenic mechanisms remain unknown.

The squamous cell carcinoma antigen recognised by T cells 3 (*SART3*) gene, also referred to as HIV-1 Tat interacting protein of 110 kDa (*TIP110*, or *p110)*, encodes an RNA binding protein that is critical to spliceosome function as it recycles small nuclear RNAs (snRNAs) during pre-mRNA splicing (see ref. 15 and references therein).

Here we describe a congenital syndrome characterised by gonadal dysgenesis in 46,XY individuals, intellectual disability, global developmental delay and overlapping neurological findings, including hypoplasia of the corpus callosum, cerebral and cerebellar anomalies. All affected individuals carry bi-allelic variants in *SART3*. We demonstrate that knockdown (KD) of the *Drosophila* orthologue of *SART3*, *Rnp4f*, causes disrupted testicular development and neuronal defects. Human induced pluripotent stem cells (iPSCs) carrying *SART3* patient variants have significant transcriptomic and proteomic differences to controls and show aberrant differentiation into gonadal and neuronal cells. Taken together our patient cohort and functional data establish bi-allelic variants in *SART3* as a cause of a spliceosomopathy affecting the male reproductive and central nervous systems.

## Results

### Bi-allelic variants in *SART3* are associated with a neuro-gonadal syndrome

Our group and collaborating laboratories identified six families with affected children sharing overlapping clinical features (global developmental delay, intellectual disability, gonadal dysgenesis in 46,XY patients) (Table 1, additional clinical information available in Supplementary Data 1) carrying bi-allelic variants in the *SART3* gene (Fig. 1a). This includes the index families, ISR1 and ISR2 (Fig. 1a) who share an ethnic origin (Moroccan Libyan Jewish, living in Israel). Causative variants in diagnostic genes for DSD including *CMA, MECP2, CDKL5, FOXG1, DAX1, L1CAM* and *ARX* were ruled out by Sanger sequencing, microarray and whole exome sequencing (WES). Analysis of WES data signified relatedness (PLINK[16] analysis, Supplementary Table 1), and just one non-synonymous variant with recessive inheritance was identified: a single nucleotide variant in *SART3* (NM_014706.3: c.2507G>A: p.Arg836Gln) (Fig. 1a and Supplementary Fig. 1a–c). This variant lies within a 1.49 cM region on chromosome 12 with a loss of heterozygosity (Supplementary Fig. 1d, e). Unaffected parents from both families and three younger unaffected siblings in ISR2 were found to be heterozygous for the *SART3* variant.

Unrelated families from Tunisia, Israel, Italy and France (TUN1, ISR3, ITA1 and FRA1) also had affected children who harboured recessive variants in *SART3* (Table 1 and Fig. 1a). In all patients, variants

in diagnostic genes were not found by WES. TUN1.1 is homozygous for a *SART3* missense variant inherited from unaffected parents who are reportedly second-degree cousins, whereas in ISR3, FRA1 and ITA1 children inherited compound heterozygous variants from unaffected non-consanguineous parents (Fig. 1a). Genetic testing was not available for unaffected siblings in FRA1 and ISR3.

The main clinical features of the nine affected individuals from the six families are summarised in Table 1. Five children carry male sex chromosomes (46,XY) and present with gonadal dysgenesis. Two of these children (TUN1.1 and ISR3.1) were born with complete 46,XY gonadal dysgenesis, presenting with no detectible testes, female external genitalia, Müllerian derivatives (uterus/vagina), and low or undetectable testosterone and Anti-Müllerian hormone (AMH) as well as normal or elevated gonadotropins (Table 1). The other 46,XY children have gonadal dysgenesis but with evidence of some testicular function/testosterone production during development as indicated by ambiguous genitalia or virilisation. At birth ISR1.1 presented with hypoplastic clitoris, fusion of the labia minora and a rudimentary phallus. Müllerian structures were not detected, and a possible streak gonad was observed on ultrasound. AMH and testosterone levels were low and the latter was unresponsive to human chorionic gonadotropin (hCG) stimulation, indicating an absence of functional testicular tissue (Table 1). ISR2.2 was born with under-virilisation including severe micropenis (1.3/0.8 cm). Hormonal evaluation at minipuberty found undetectable basal testosterone, a poor response to hCG stimulation and undetectable AMH, indicating primary hypogonadism (gonadal dysgenesis). Secondary hypogonadism was also suggested by the lack of elevated gonadotropins at this time (Table 1). Both an abdominal left testis and a palpable inguinal right testis were resected during childhood with histology revealing abundant fibrotic tissue (Fig. 1b–g). Epididymis (Fig. 1c, f) and fallopian tubes (Fig. 1d, g) were found adjacent to both gonads. Ultrasound also revealed a small uterus and vagina. TUN1.2 also had under-virilisation at birth (micropenis, impalpable testes). No Müllerian structures were identified and left and right epididymides were present. FSH was elevated (Table 1). A diagnosis of partial 46,XY gonadal dysgenesis was made.

Unlike 46,XY children, 46,XX individuals with bi-allelic *SART3* variants were born with typical female external genitalia and there is no evidence supporting a reproductive disorder. ISR1.2 had pubic hair and thelarche but had not yet reported menstruation at age 16. ISR2.1 had ovaries present on ultrasound and underwent normal pubertal development with menstruation reported at age 12.5. ITA1.1 and FRA1.1 have no reported endocrine disruptions and are pre-pubertal (Supplementary Data 1).

Neurodevelopmental comorbidities were noted in all affected individuals. All children had intellectual disability (ID) reported, ranging from mild to profound, and hypotonia. All were reported to have global developmental delay with most showing delays in all aspects including motor skills, speech and language, communication and socioemotional skills (Table 1). All individuals had craniofacial anomalies reported. Common neurological observations included corpus callosal agenesis (Fig. 1h) or hypoplasia (e.g. Fig. 1j, l, n, p, s, u) (8/9 assessed). Cerebral anomalies were found in all but one child including ventriculomegaly (Fig. 1i, m, o, t, v). Cerebellar abnormalities were found in 6 children including vermian atrophy (Fig. 1h, l, n, s, u). White matter abnormalities (e.g. delayed myelination) were reported in five children (Fig. 1i, k, m, o and Table 1) and four individuals had seizures or epileptiform activity reported on EEG (Table 1). FRA1.1 had pons hypoplasia (Fig. 1s) and low choline was reported on MR spectroscopy (additional MRI images are shown in Supplementary Fig. 1f–t). Three children had cardiac defects reported.

In summary, children born with recessive variants in the *SART3* gene present with a hitherto-undescribed syndrome characterised by gonadal dysgenesis specific to the testis (46,XY gonadal dysgenesis), neuro-developmental defects and intellectual disability.

**Table 1 | The main clinical findings for the nine children**

| Patient ID | ISR1.1[a] | ISR1.2 | ISR2.2[a] | ISR2.1 | TUN1.1[a] | TUN1.2 | ISR3.1[a] | ITA1.1[a] | FRA1.1[a] |
|---|---|---|---|---|---|---|---|---|---|
| SART3 variant/s | c.2507G>A: p.R836Q (hom.) | c.2507G>A: p.R836Q (hom.) | c.2507G>A: p.R836Q (hom.) | c.2507G>A: p.R836Q (hom.) | c.631G>A: p.E211K (hom.) | c.631G>A: p.E211K (hom.) | c.2299C>T: p.R767W c.757C>T: p.R253[a] | c.1477C>T: p.R493W c.646T>C: p.S216P | c.2153C>T: p.P718L c.1555A>G: p.R519G |
| Karyotype | 46, XY | 46, XX | 46, XY | 46, XX | 46, XY | 46, XY | 46, XY | 46, XX | 46, XX |
| Sex of rearing | Female | Female | Male | Female | Female | Male | Female | Female | Female |
| Age at exam | 5.5yo; 10yo; 12yo | 3.5yo; 8.5yo; 10yo | 3yo; 5yo | 4yo | 16mo; 4.5yo | 4mo; 1yo, 2.5yo | 7mo; 2.75yo | 7yo | 1mo, 7mo, 7yo |
| Ges. age and weight | 35 wks, 2.9 kg | FT, 3.12 kg | FT, 3.3 kg | FT, 4.6 kg | FT, 3 kg | FT, 3.5 kg | 34 + 5 wks, 2.16 kg | FT, 2.75 kg | FT, 3.2 kg |
| Facial dysmorphism | Long face, metopic ridge, sparse eyebrows, bulbous nasal tip, high arched palate | Long face, metopic ridge, sparse eyebrows, bulbous nasal tip, high arched palate | Supressed nasal bridge, microretrognatia, high arched palate | Broad nasal bridge, deep filtrum | Bilateral strabismus | High forehead, broad nose, long philtrum, thin upper lip, low/posterior rotation | Deep-set eyes | Upslanting palpebral fissures, thin lips, flat philtrum | Deep-set eyes, bulbous nasal tip, thin upper lip, large flat philtrum |
| Microcephaly | Borderline | Borderline | – | – | + | – | Borderline | – | Borderline |
| Neurological findings | | | | | | | | | |
| Intellectual disability | Severe | Severe | Mild-mod | Mild-mod | Severe | Yes | Severe | Moderate | Profound |
| Developmental delay | GDD (severe) | GDD (severe) | GDD (mild-mod) | GDD (mild) | GDD | GDD | GDD | GDD | GDD |
| Hypotonia | + | + | Mild | Mild | + | + | + | Mild | + |
| Spasticity | Mild | Mild | – | – | – | – | – | – | + |
| Ataxia | Not walking | Not walking | Mild | – | – | – | N/A | Mild | Not walking |
| Abnormal movement | + | + | – | – | – | – | – | + | + |
| Irritability/Sleep Disorder | + | + | – | – | – | – | – | – | + |
| Autistic behaviour | + | + | – | – | – | – | – | – | + |
| Seizures | + | –/EA | – | –/EA | – | – | + | – | – |
| Cerebral | Colpocephaly. Acute hydrocephalus | Ragged PLV configuration | Mild VM. Ragged PLV configuration | Mild VM. Ragged PLV configuration | Moderate vermian atrophy | MRI normal at 1yo | VM | Mild VM | Mild VM |
| Corpus callosum | Agenesis | Thin | Thin | Thin | Thin | – | Thin | Thin | Thin |
| White matter | DM and atrophy | DM and atrophy | DM and atrophy | DM and atrophy | – | – | DM | – | N/A |
| Cerebellar | Mild CVA | – | Mild CVA | Mild CVA | CVA | N/A | N/A | Mild CVA | CVA, HA, thin BS |
| Genitourinary findings | | | | | | | | | |
| External genitalia | Hypoplastic clitoris, labia minora fusion, rudimentary phallus | Female appearance | Severe micropenis, hypoplastic scrotum | Female appearance | Single genitourinary orifice | Micropenis | Female appearance | Female appearance | Female appearance |
| Gonads | Streak/GD? | N/A | PGD/regression | Ovaries | Not palpable. GD? | GD after ochidopexy | Undetected. GD? | N/A | N/A |
| Müllerian structures | – | + | +/Fallopian tubes, small uterus and vagina | + | +/Small uterus and vagina | – | Uterus | N/A | N/A |
| Wolffian structures | – | – | Epididymides | – | – | Epididymides | – | N/A | N/A |

**Table 1 (continued) | The main clinical findings for the nine children**

| Patient ID | ISR1.1[a] | ISR1.2 | ISR2.2[a] | ISR2.1 | TUN1.1[a] | TUN1.2 | ISR3.1[a] | ITA1.1[a] | FRA1.1[a] |
|---|---|---|---|---|---|---|---|---|---|
| Endocrinological findings | | | | | | | | | |
| Basal gonadotropins | N/A | N/A | LH - U at 2do, 2mo. FSH - U at 2do, NL at 2mo | N/A | N/A | FSH - EL at 2.5yo | LH - NL at 16mo, 17mo. FSH - EL at 17mo | N/A | N/A |
| Basal androgens | Low | N/A | NL at 2do, U at 2mo | N/A | U at 16mo and 4.5yo | U at 2.5yo | T - U at 17mo. Androstendione - NL at 17mo | N/A | N/A |
| hCG stimulation test | Unresponsive | N/A | Low response at 2mo | N/A | N/A | N/A | N/A | N/A | N/A |
| AMH | Low | N/A | U at 2do, 2mo | N/A | U at 16mo | N/A | N/A | N/A | N/A |

Summary of the main clinical findings including neurological features and their manifestations, genitourinary, gonadal, and endocrinological findings. Further clinical details can be found in the Supplementary Data 1 table.
*N/A* not available or unknown, *hom* homozygous, *do* day old, *yo* year old, *FT* full term, *wks* weeks, *GDD* global developmental delay, *EA* epileptiform activity, *PLV* posterior lateral ventricles, *VM* ventriculomegaly, *MRI* magnetic resonance imaging, *DM* delayed myelination, *CVA* cerebellar vermis atrophy, *HA* hemisphere atrophy, *BS* brainstem, *PGD* partial gonadal dysgenesis, *GD?* suspected gonadal dysgenesis, *GD* gonadal dysgenesis, *U* undetectable, *AMH* Anti-Müllerian Hormone, *LH* luteinizing hormone, *FSH* follicle stimulating hormone, *T* testosterone, *IU/L* international units per L, *hCG* human chorionic gonadotropins, *T* testosterone, *PTH* parathyroid hormone, *NL* normal levels, *EL* elevated.
[a]Indicates the probands. A negative symbol (−) denotes no phenotype, a positive symbol (+) denotes the presence of a phenotype.

## SART3 variants fall in highly conserved protein domains

All *SART3* variants are absent or extremely rare in public variant databases (dbSNP138, 1000 Genomes, NHLBI GO Exome Sequencing Project, ExAC, GnomAD, Iranome; Supplementary Table 2). The variant allele c.2507G>A: p.Arg836Gln (families ISR1/2) was not found in a control cohort of 165 healthy women of Moroccan Jewish ancestry. None of the *SART3* variants were found in a large WES database of Israeli probands and controls tested at the Wolfson Medical Center, or in 800 individuals of North African origin. In-silico tools such as Polyphen-2, PROVEAN, SNAP2, muPRO, SIFT, predict variants to be mostly pathogenic/damaging (Supplementary Table 2). *SART3* demonstrates Loss of Function (LOF) constraint (pLI = 1) and while nine *SART3* LOF variants are reported in GnomAD (v2.1.1) they are absent as homozygous.

Human *SART3* is a gene with 22 exons encoding a protein of 963 amino acids (Fig. 2a). All patient variants affect highly conserved amino acids in protein domains essential for SART3 function (Fig. 2a). Five variants fall within the 12 half-a-tetratricopeptide (HAT) repeats domain necessary for protein-protein interactions and in vitro splicing activity[15]. Variant p.Arg519Gly affects a residue implicated in *SART3* dimerisation[17] (Fig. 2b) and p.Arg493Trp affects a residue involved in binding to the ubiquitin-specific proteases USP15 and USP4[17,18] (Fig. 2c). Ser216 is predicted to be phosphorylated (PhosphositePlus). The Arg253* variant in patient ISR3.1 is predicted to cause a premature termination within the HAT domain (Fig. 2a). Three variants affect highly-conserved residues in the RNA recognition motifs (RRM1 or RRM2) (Fig. 2a), domains which play a role in the recognition and binding of RNA and in vitro splicing activity, namely U6 snRNA and spliceosome recycling[19]. SART3 has been shown to localise to both the cytoplasm and the nucleus; in the latter it is dispersed throughout the nucleoplasm, including in Cajal bodies, but is excluded from nucleoli (see[15] for a review). To test variants for potential mislocalisation or altered protein stability, FLAG-tagged SART3 variants were transiently transfected into HEK293t cells. Western blot analysis found a significant loss of signal for the p.Arg253* variant. In addition, p.Ser216-Pro, p.Arg493Trp, p.Arg519Gly and p.Pro718Leu variants had a reduced signal with FLAG and SART3 antibodies, although this did not reach statistical significance over three replicates (Fig. 2d, e and Supplementary Fig. 2a, b). Immunostaining showed that all missense variants demonstrated nuclear localisation similar to endogenous SART3 and transiently expressed WT SART3 (Fig. 2f). Additional western blotting and immunofluorescence examples can be found in Supplementary Fig. 2.

Given the strong clinical and genetic evidence that *SART3* underlies the conditions described here, we then sought to analyse the functional implications of variants in *SART3* in a range of model systems, namely *Drosophila*, mouse and human pluripotent stem cells.

## SART3 has a conserved role in *Drosophila* gonadal and neural development

The role of SART3 in assembly of the major spliceosome is conserved from yeast to humans[15]. In *Drosophila* embryos expression of the *SART3* orthologue, *RNA- binding protein 4F* (*Rnp4f*, Unipro ID: Q9W4D2) is highest in the gonads, central nervous system (ventral nerve cord and brain) and gut (Supplementary Fig. 3a–d)[20]. To assess the role of *Rnp4f* during *Drosophila* development we used RNAi-mediated knockdown (KD), as poor protein conservation between human SART3 and Rnp4f (20% identity, 34% similarity) precluded us from introducing patient variants. Global KD using the inducible *tubulin*-GAL4/upstream activation sequence (UAS) system to drive the RNAi transgene resulted in a 70% reduction of *Rnp4f* (Supplementary Fig. 3f). This caused significant lethality, with widespread death at the embryonic and pupal stages and only half the expected survival to adulthood (Supplementary Fig. 3f). This is consistent with previous studies demonstrating that global loss of *Rnp4f* results in embryonic

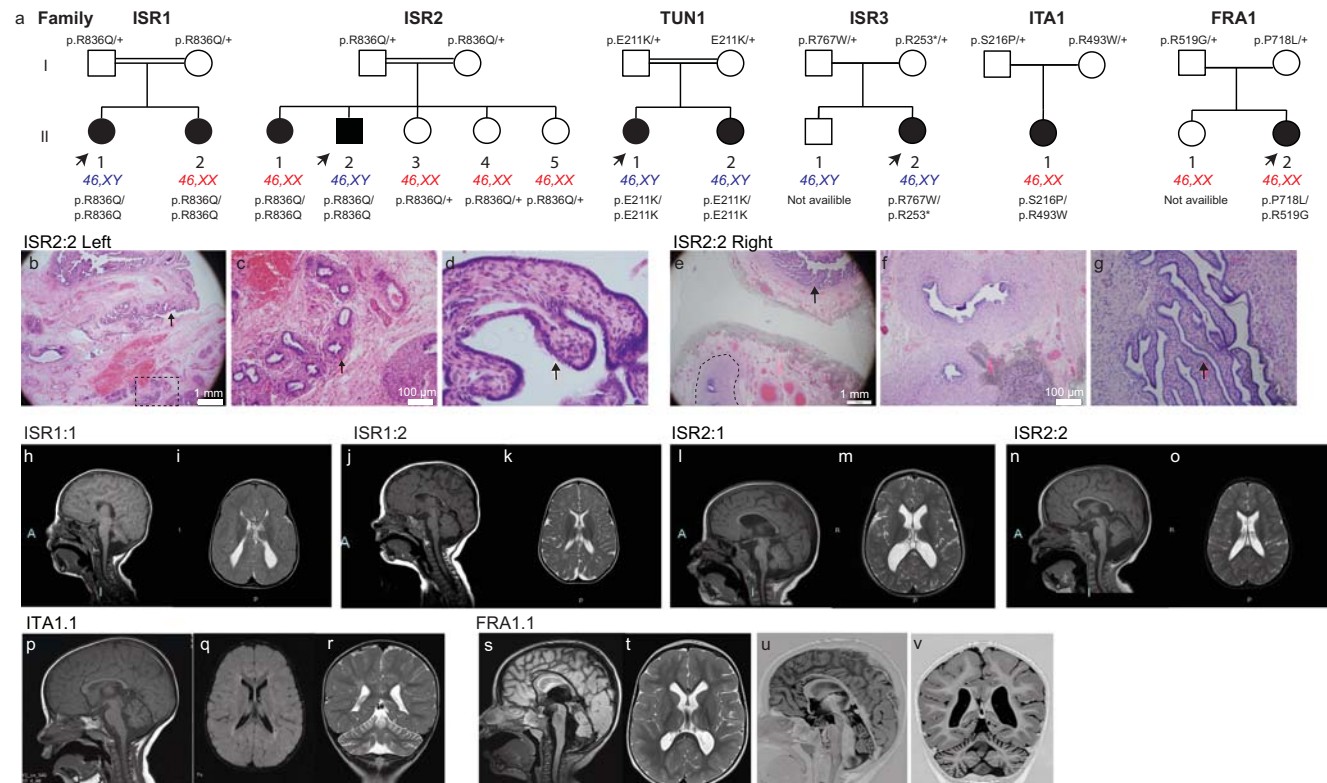

**Fig. 1 | Bi-allelic variants in *SART3* are associated with a syndrome characterised by developmental delay, intellectual disability and 46,XY-specific gonadal dysgenesis. a** The pedigrees of six families with affected children carrying bi-allelic *SART3* variants. Arrows = probands. Both genetic sex (XX or XY) and gender (circle = female, square = male) are indicated. Families ISR1, ISR2 and TUN1 have homozygous variants. Families ISR3, ITA1, FRA1 have compound heterozygous variants. **b**–**g** Gonad histology in ISR2.2. **b** The left testis, resected at 9 months, was fibrotic. Both epididymis (box) and fallopian tube (arrow) were present. **c** Enlarged views of the left epididymis (insert from **b**) and **d** fallopian tube (arrow). **e** Histology of the right testis, resected at age 5, demonstrated widespread fibrosis and the presence of epididymis (dotted line) and fallopian tube (arrow). **f** Higher magnification of the epididymis, and **g** the fallopian tube. **h**–**v** MRI imaging. T1-weighted midline sagittal images demonstrating thin corpus callosum in **j** ISR1.2, **l** ISR2.1, **n** ISR2.2, **p** ITA1.1 and FRA1.1 (**s** and **u**, inversion recovery image 0.9 mm thick), and **h** absent corpus callosum in ISR1.1. Atrophy of the cerebellar vermis is observed in **h** ISR1.1, **l** ISR2.1, **n** ISR2.2 and in **p** ITA1.1. **s**, **u** In FRA1.1, sagittal imaging also revealed vermis atrophy, enlargement of the 4th ventricle, cisterna magna and hypoplastic pons. T2-weighted axial images at the level of the lateral ventricles demonstrated delayed myelination and atrophy of the white matter and ragged configuration of the posterior horn of the lateral ventricle in **k** ISR1.2, **m** ISR2.1 and **o** ISR2.2. **r** Coronal section for ITA1.1 shows cerebellar atrophy with **q** axial inversion recovery image showing mild ventriculomegaly. **i** ISR1.1 has colpocephaly with decreased white matter volume. **t** Axial T2-weighted image in FRA1.1 demonstrates ventriculomegaly with square-shaped frontal horns. **v** Atrophy of the cerebellar hemispheres is also observed in a coronal inversion recovery image. A anterior.

lethality[20,21]. KD embryos had severe CNS defects (Fig. 3a, b). Surviving KD adults had held-out wings, proboscis defects and generalised atrophy (Supplementary Fig. 3g–j) and males were infertile. Immunofluorescent staining of testes from these KD males (Fig. 3f–h) demonstrated a loss of round spermatocytes and spermatid populations, identified by the absence of cells with reduced Vasa expression (Fig. 3f –arrows) compared to controls (Fig. 3c). Late-stage spermatid bundles, visualised with DAPI in controls (Fig. 3c–e), were also absent in KD testes (Fig. 3g, h). It therefore appears that *Rnp4f* is required for spermatogenesis. Testicular somatic cells offer essential support to the germline during spermatogenesis in flies and humans. Gonadal dysgenesis phenotypes in 46,XY patients suggest a role for SART3 in the somatic cells of the testis, as disruption in the germline alone would not result in gonadal dysgenesis. To assess whether Rnp4f is functioning in the somatic cells in *Drosophila*, RNAi transgenes were expressed using the somatic cell-specific driver *traffic jam*-Gal4 (*tj*-Gal4). This resulted in immature, misshaped testes (Fig. 3k, l) with changes in spermatogenesis similar to those found in the global KD flies. Phase contrast imaging suggested an accumulation of early germ cells/spermatocytes, and very few motile sperm in the somatic KD testis (Fig. 3k), compared to controls (Fig. 3i). Immunofluorescence using Don Juan-GFP, a marker of elongated spermatids, found these to be reduced in the KD (Fig. 3j, l). Thus, SART3 appears to have a

conserved role in *Drosophila* testis development, where it is required in the somatic gonadal cells for proper spermatogenesis.

Since all affected patients have a range of neurodevelopmental comorbidities, we also investigated the role of *Rnp4f* in embryonic neurons and glia in the *Drosophila* model. Neuronal-specific KD (using the *elav*-Gal4 driver) caused complete embryonic lethality due to disrupted neuronal development, including holes along the midline (Fig. 3n). Reducing RNAi expression by raising embryos at a lower temperature resulted in less severe defects, demonstrating the phenotype severity was RNAi-expression dependent (Fig. 3o). Importantly, KD exclusively in the glia (using the *repo*-Gal4 driver) had no discernible effect (Fig. 3p, q). Our findings therefore demonstrate that *Rnp4f* is required for normal neural development during embryogenesis in *Drosophila*. Additional examples including those with a second RNAi transgene can be found in Supplementary Fig. 3. While the *Drosophila* data were compelling, and consistent with the clinical presentation of patients, we then sought to analyse the functional impact of *SART3* gene variants in a mammalian system using mouse models.

### Attempts to generate *SART3* variant knockin mouse models were unsuccessful

Early lethality associated with loss of *SART3* in numerous animal models including zebrafish[22] and mice[23] has hindered an

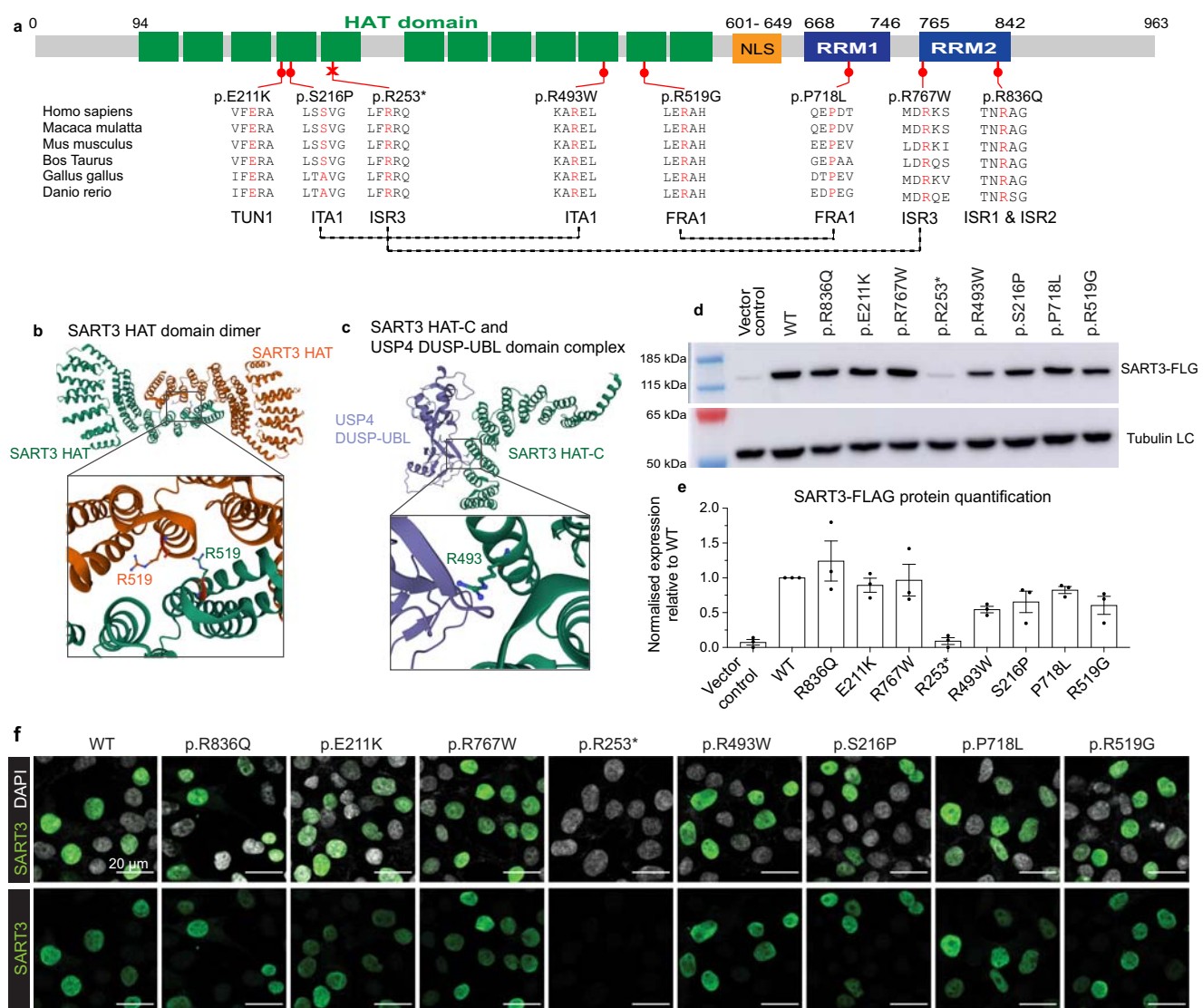

**Fig. 2 | SART3 variants map to conserved domains and residues. a** SART3 is a 963 amino acid protein containing 12 half-a-tetratricopeptide (HAT) repeats (the HAT domain, green), a nuclear localisation signal (NLS, yellow) and two RNA recognition motifs (RRM1 and RRM2, blue). Positions of SART3 variants are shown in red. Variants show amino acid conservation between vertebrate species. **b** Crystal structure of human SART3 HAT domains as a dimer (PDB ID 5CTQ) with Arginine 519 highlighted on each chain. **c** Crystal structure of human SART3 HAT-C domain-human USP4 DUSP-UBL domain complex (PDB ID 5CTR) with Arginine 493 highlighted. Crystal structures were visualised in the RCSB PDB 3D protein viewer (Mol*). **d** Western blot analysis of WT and variant pCMV-SART3-FLAG constructs transiently transfected in HEK293t cells with an anti-FLAG antibody and beta-tubulin loading control (LC). **e** Quantification of western blot SART3-FLAG (FLG) levels, normalised to the beta-Tubulin LC and relative to WT. Data is represented as mean ± SEM from n = 3 independent experiments. Significant was calculated using a one-way ANOVA with Dunnett's multiple comparison (P values = *<0.05; **<0.01; ***<0.001). The p.R253* variant shows a significant drop in expression (P value = 0.001). Raw data and uncropped blots are provided in a source data file. **f** Immunofluorescence analysis of WT and variant pCMV-SART3-FLAG constructs transiently transfected in HEK293t cells using a SART3 antibody. WT and missense variants showed nuclear localisation. The truncating variant (p.R253*) was not detected with the SART3 or FLAG antibody (Supplementary Fig. 2).

understanding of its role during embryonic development, with attempts to create conditional mouse knockouts proving unsuccessful[23]. We endeavoured to create mice carrying the *SART3* variants found in families ISR1 and 2 (p.Arg836Gln) and TUN1 (p.Glu211Lys) via CRISPR/Cas9 genome editing, as these were the first variants identified, fell in separate protein domains and had strong genetic evidence for causation. Despite repeated attempts at multiple facilities we were unable to generate correctly-targeted knockin mice (either heterozygous or homozygous; Supplementary Table 3). In total, 14 injection sessions of ~3000 embryos yielded 51 pups, all of which were WT except for 10 which carried heterozygous insertion/deletions (INDELs) at the target site. Breeding showed these null alleles were homozygous

lethal, consistent with previous work[23]. We therefore sought additional human cell-based models in which to investigate patient variants.

**Human induced pluripotent stem cells carrying *SART3* patient variants show disrupted differentiation into testis and neural cell lineages**

To confirm the pathogenicity of the patient variants during human development we used CRISPR/Cas9 gene editing to generate iPSCs heterozygous or homozygous for the p.Arg836Gln variant (Supplementary Fig. 4). This variant was prioritised as it had robust genetic data for causation from two families and affects a well-established domain essential for SART3 function. Edited lines had extensive quality

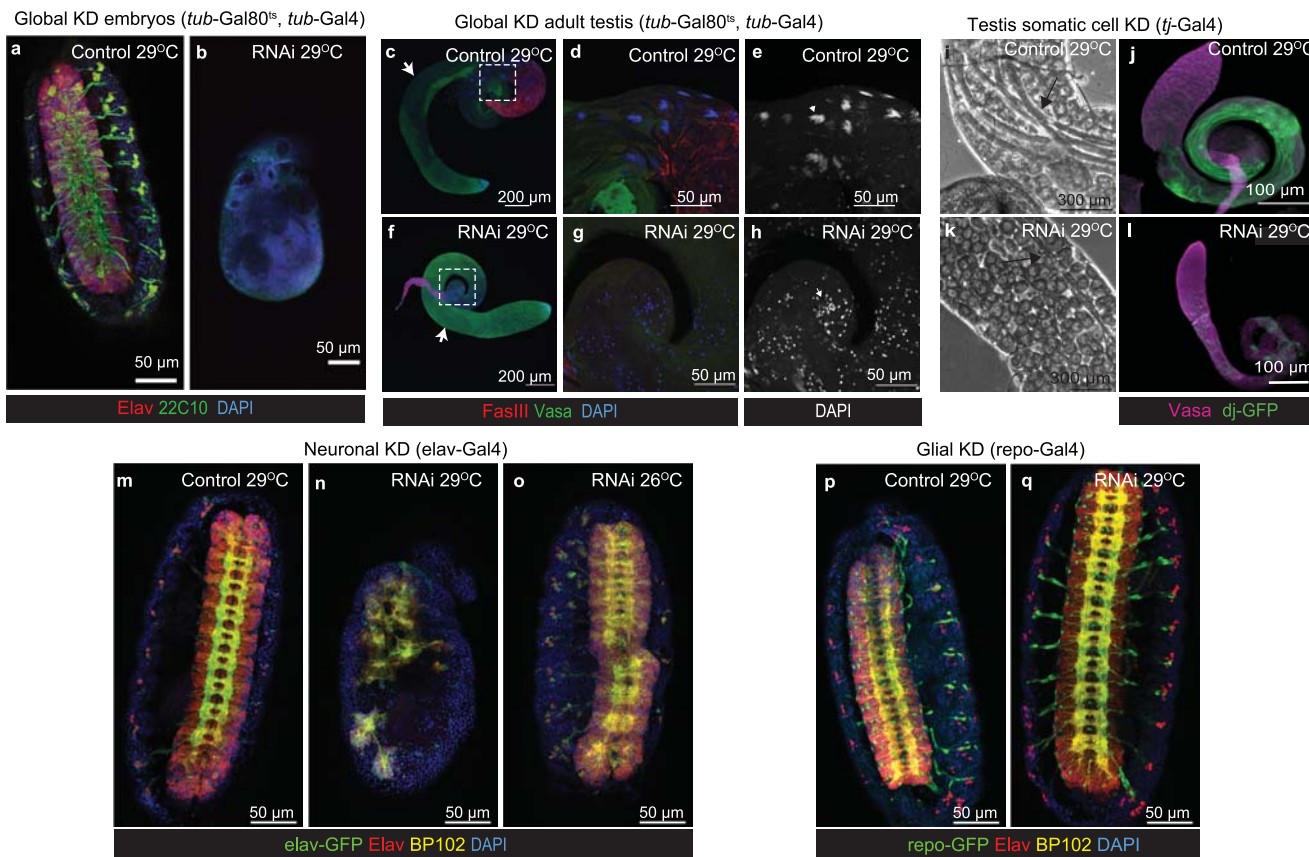

**Fig. 3 | SART3 is a conserved regulator of embryonic CNS and testis development. a**, **b** Stage 16 *Drosophila* embryos stained with CNS markers Elav (red; pan-neuronal), 22C10 (green; neuronal cell bodies and axons), and DAPI (blue; nuclei). **a** *w*[1118] crossed control embryos. **b** Global *Rnp4f* KD using the *tubulin*-Gal4; *tubulin*-Gal80 temperature sensitive driver (*tub*-Gal4, *tub*-GAL80[ts]) to express RNAi at the permissive temperature of 29 °C. Significant embryonic lethality and disrupted CNS development was observed. **c**–**e** Control and **f**–**h** *tub*-Gal4, *tub*-GAL80[ts], RNAi adult male testes stained for Vasa (green; germ cells), Fasciclin 3 (FasIII; red; hub) and DAPI (blue; spermatid bundles). **c** Vasa stains germs cells from stem cell to spermatogonia stages and is strongest in earlier germ cell populations with round spermatocytes and spermatids visualised by reduced intensity (arrow). **f** Testes from *tub*-Gal4, *tub*-GAL80[ts], RNAi males have uniform Vasa expression, suggesting a loss of round spermatocytes and spermatids (**f**, arrow). **d**, **e** Higher magnification of insert from **c** showing DAPI-positive spermatid bundles in the control (arrowhead). **g**, **h** Higher magnification of insert from **f** showing KD testes had no spermatid

bundles. **i**–**l** Day 14 testis from **i**, **j** controls and **k**, **l** *Rnp4f* KD using a testicular somatic cell driver, *traffic jam*-Gal4 (*tj*-Gal4). **i** Phase contrast image showing spermatid bundles in the control (arrow). **j** Control testis expressing the Don Juan-GFP reporter for elongated spermatids (green) and stained for Vasa (magenta; germ cells). **k** Somatic cell KD testes accumulate earlier germ cell stages and have less spermatid bundles. **l** They also lose elongated spermatids. **m**–**o** Stage 16 *Drosophila* embryos expressing *elav*-GFP (green; pan-neuronal marker) stained with anti-Elav (red), anti-BP102 (yellow; axons) and DAPI (blue). **n** Neuronal-specific KD (*elav*-Gal4, UAS-GFP) led to embryonic lethality and midline CNS defects at 29 °C compared to **m** control. **o** Less severe defects were observed at 26 °C, where RNAi expression is lower. **p**, **q** Stage 16 *Drosophila* embryos with glial-specific expression (*repo*-Gal4, UAS-GFP) stained for Elav (red), BP102 (yellow) and DAPI (blue). **p** No embryonic defects were found in *w*[1118]-crossed control or **q** KD at 29 °C. **b**, **f**–**h**, **n**, **o**, **q** are examples of KD using RNAi-B and **k**, **l** are using RNAi-V. Additional examples provided in Supplementary Fig. 3.

control testing including flow cytometry and embryoid body formation to confirm pluripotency (Supplementary Fig. 4).

*SART3* mRNA is expressed in the human embryonic gonad throughout development (weeks 4 to weeks 25 gestation, Reprogenomics viewer[24,25]). In human foetal testis (9 + 2 gestational weeks, GW) SART3 is present in speckles within the nucleus and lower levels in the cytoplasm of Sertoli cells, germ cells and interstitial cells (Fig. 4a–d). Staining was also observed in the foetal ovary (Supplementary Fig. 4z–ac). To assess whether a patient variant in *SART3* can cause disruption to human testis development, we differentiated iPSCs into testis-like organoids[26] (Fig. 4e). Despite using a procedure in which the same number of iPSCs were used to establish each differentiation at day 0, and for organoid assembly at day 7, day 14 and day 21 testis-like organoids from the homozygous variant line were significantly smaller than controls (day 14 organoid diameter: control $x = 3742 \pm 441.3$ mm, homozygous p.Arg836Gln variant line $x = 2689 \pm 464.7$ mm, $P$ value < 0.001; day 21 control $x = 3531 \pm 690$ mm, homozygous p.Arg836Gln variant line $x = 2625 \pm 490.1$ mm, $P$ value = 0.002; Fig. 4f).

Notwithstanding size differences, organoids from all three cell lines had tubular structures expressing SOX9 (Fig. 4g, m, s, k, q, w), GATA4 (Fig. 4h, n, t) and Collagen IV (COLIV Fig. 4j, p, v). Greater levels of apoptosis (cleaved Caspase-3, yellow) were observed in the homozygous variants compared to the heterozygous and the controls (Fig. 4l, r, x). Higher magnification shows cleaved Caspase-3 staining affecting cells outside tubular structures (Fig. 4x). Gene expression analysis for markers during the differentiation showed similar trends for all cell lines wherein *OCT4* (pluripotency) reduces from day 0, *PAX2* (intermediate mesoderm) is elevated at day 7 and then reduces as bi-potential gonadal markers *GATA4* and *NR5A1* are activated. Sertoli cell markers including *SOX9* and *CLDN11* are activated from day 7, increasing up to day 21. Notably, the *SART3* variant lines showed some differences to the controls including increased *OCT4* expression at day 0 and reduced *PAX2* activation at day 7. *GATA4* expression was higher at all timepoints after day 7 in the variant lines. Variant cells also failed to upregulate *NR5A1* and *FGF9*, although *SOX9* and *CLDN11* expression was not disrupted. Taken together these results demonstrate that a

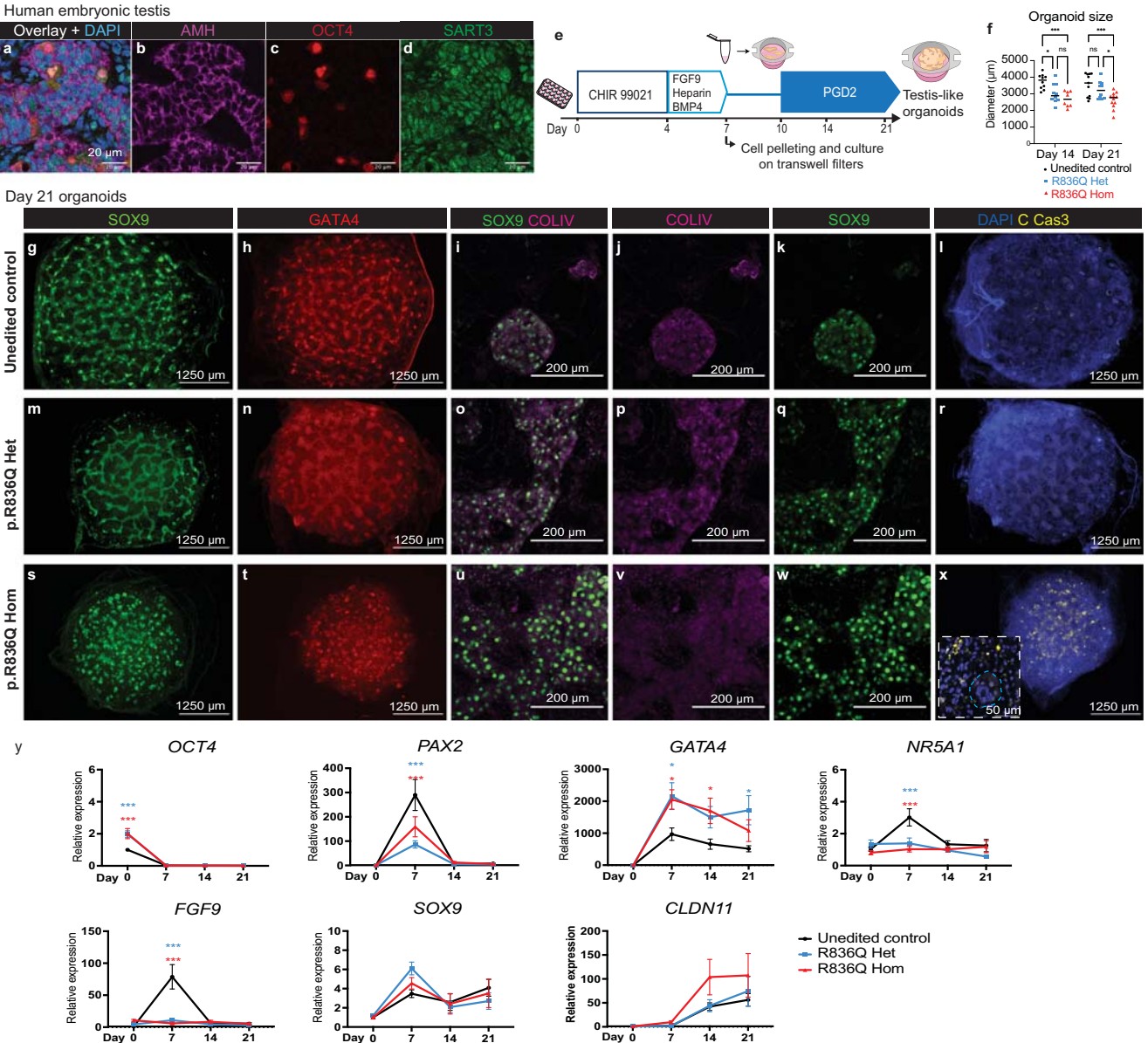

**Fig. 4 | SART3 is expressed in human foetal gonads, and patient variants lead to disrupted differentiation into gonad-like organoids. a–d** 9 weeks + 2 days gestation human embryonic testis stained for **b** AMH (Sertoli cells, magenta) and **c** OCT4 (germ cells, red). **d** SART3 (green) has speckled nuclear and cytoplasmic expression in all cell types. **a** is an overlay with DAPI (nuclei, blue). **e** Method for gonadal-like organoid differentiation. **f** Organoid size (diameter) at day 14 and day 21. Individual data points are from organoids from three separate differentiations, with mean represented by a dash. *n* = 10 for day 14 control, day 14 heterozygous, day 21 heterozygous. *n* = 8 for day 14 homozygous. *n* = 11 for day 21 control and day 21 homozygous. Significance was calculated using one-way ANOVA with Tukey's multiple comparison test. *P* values = *< 0.05; **<0.01; ***<0.001. Homozygous *SART3* variant organoids were smaller at both timepoints (day 14 *P* < 0.001, day 21 *P* = 0.002). **g–x** Immunostaining of day 21 organoids from **g–i** control iPSCs, **m–r** heterozygous SART3 p.Arg836Gln or **s–x** homozygous SART3 p.Arg836Gln iPSCs. **g, m, s** Testis Sertoli cell marker SOX9 (green) was observed in tubular

structures in all lines. **h, n, t** Gonadal marker GATA4 (red) was also expressed. **i–k, o–q, u–w** Higher magnification images show SOX9 (green) expressing cells inside Collagen IV (COLIV, magenta) positive tubular structures. **l, r, x** Cleaved Caspase-3 staining (apoptosis, yellow) was elevated in the **x** homozygous variant organoids compared to **l** control or **r** heterozygous organoids. An insert shows CCas-3 staining outside of tubular structures (**x**, blue dashed line). **y** RT-qPCR of gene markers during the differentiation. *OCT4* is a pluripotency marker. *PAX2* is a marker of the intermediate mesoderm. *NR5A1* and *GATA4* are bi-potential gonad markers. *FGF9, SOX9* and *CLDN11* are Sertoli cell markers. At each timepoint *n* = nine wells or organoids from three independent differentiations. Data presented as mean ± SEM relative to day 0 control. A two-way ANOVA followed by Tukey's multiple comparison test was performed to compare cell lines at each timepoint. Asterisk represents a significant difference from control line (*P* values = *<0.05; **<0.01; ***<0.001).

patient variant in *SART3* causes disruption to the differentiation of iPSCs into gonadal-like lineages.

*SART3* is expressed throughout the human adult brain, with highest expression in the cerebellar hemisphere and the cerebellum (GTeX and Human Protein Atlas). We investigated neuronal differentiation in the iPSC control and *SART3* variant lines. Directed

differentiation of unedited control iPSCs using the NGN2-lentiviral method (Fig. 5a)[27] resulted in a dense network of neurons after 14 days (Fig. 5b–f). These express the axonal marker Neurofilament (intermediate filaments; Fig. 5b, c), and the dendrite marker Microtubule associated protein 2 (MAP2, Fig. 5b, c). Ankyrin-G (axon initial segment, Fig. 5d), Beta-3-tubulin, a marker of axons, dendrites and soma

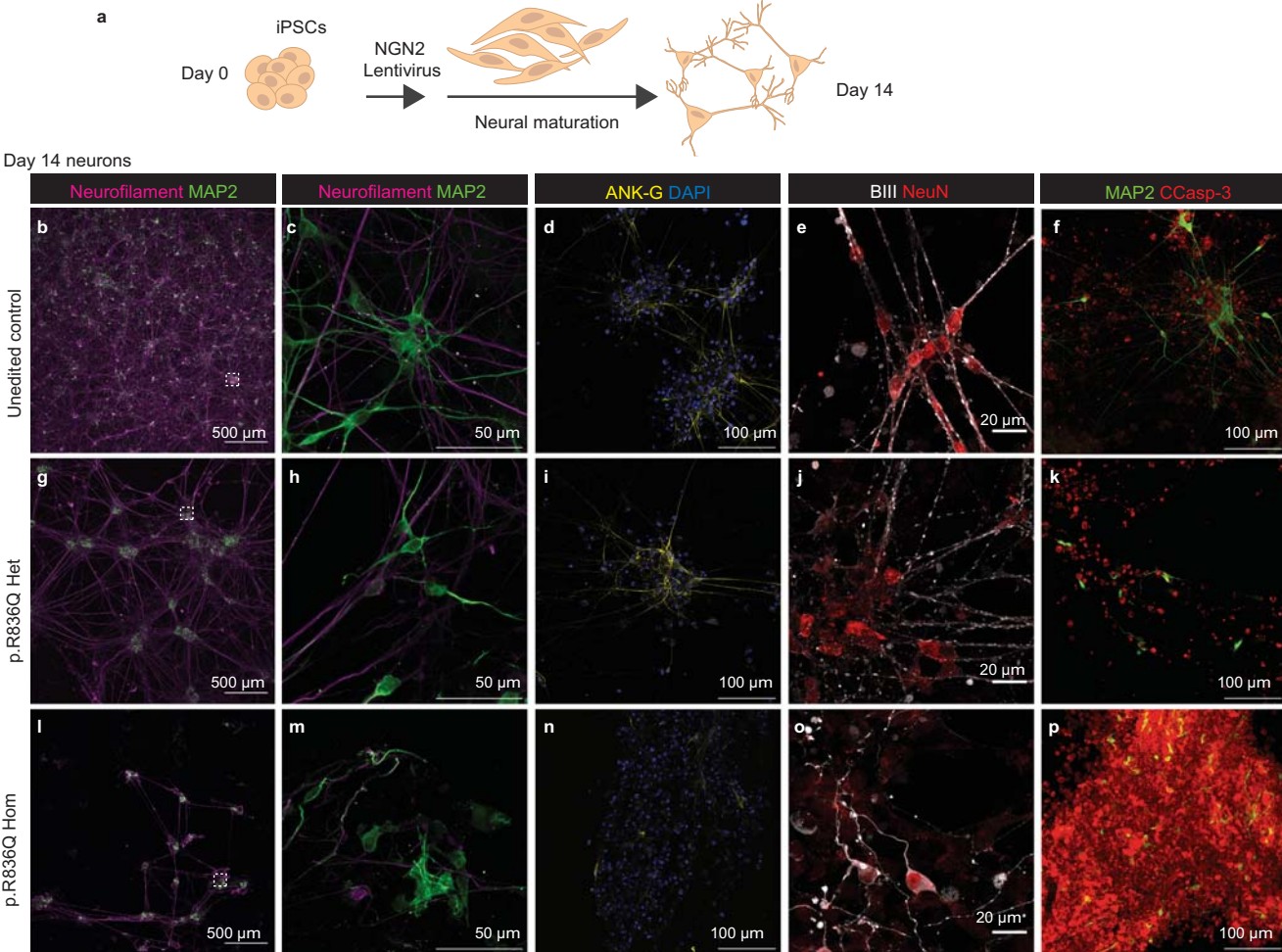

**Fig. 5 | iPSCs carrying SART3 patient variant p.Arg836Gln show disrupted differentiation into neurons in vitro. a** Overview of NGN2 protocol for differentiation of iPSCs into cortical neurons. **b–p** Day 14 neurons stained for various neuronal markers. **b, g, l** Neurofilament staining (axons, magenta) and MAP2 (dendrites, green) staining illustrates networks, with higher magnification of inserts shown in **c, h, m. d, i, n** Ankyrin-G (ANK-G, axon initial segment, yellow) and DAPI (blue). **e, j, o** β-III tubulin (BIII, axons dendrites and soma, white) with NeuN (neural soma, red). **f, k, p** MAP2 (dendrites, green) and cleaved Caspase-3 (CCasp-

3, apoptosis, red). **b–f** Neurons derived from control unedited iPSCs and **g–h** heterozygous SART3 p.Arg836Gln iPSCs form dense networks) and show expression of **b, c, g, h, f, k** MAP2, **d, i** ANK-G, **e, j** BIII and NeuN. **l–p** Homozygous SART3 p.Arg836Gln neurons are sparser, **n** show reduced expression of ANK-G and have **m, o** disrupted morphology. **f, k, p** Cleaved Caspase-3 staining indicates increased apoptosis in the **p** homozygous variant line compared to the **f** control or **k** heterozygous line.

(BIII; Fig. 5e), and NeuN, which marks the neural soma, were also expressed (Fig. 5e). The iPSCs carrying a heterozygous p.Arg836Gln variant demonstrated efficient neuronal differentiation but formed less dense neural networks (Fig. 5g–k). In contrast, the homozygous p.Arg836Gln variant iPSC cells showed markedly disrupted differentiation (Fig. 5l–p) with very few mature neurons detected (Fig. 5l). In these neurons very little Ankyrin-G expression was observed, and other markers highlighted impaired neural maturation and dysmorphic morphology including absent or disrupted axonal projections (Fig. 5m–o). Greater apoptosis (cleaved Caspase-3) was observed in the homozygous cell line compared to the heterozygous and control lines (Fig. 5f, k, p). Given the reduced capacity of *SART3* variant iPSC lines to differentiate into gonadal or neuronal lineages, we turned to transcriptomic and proteomic analysis of undifferentiated iPSCs to investigate the patho-molecular mechanisms involved.

### *SART3* variant iPSCs have widespread changes to gene expression and signalling

Significant differences were observed in the RNA and protein expression profiles between the unedited control, heterozygous and homozygous SART3.p.Arg836Gln iPSC lines, with clustering of the biological triplicates within each genotype (Supplementary Fig. 5a, b). RNA-seq analysis found 295 genes differentially expressed (DE) with a false discovery rate (FDR) < 0.05, between all three groups. The biggest differences were observed between the unedited control cells and the homozygous variant line (Fig. 6a), where 2487 genes were downregulated and 2278 upregulated in the variant cells. KEGG pathway and Gene Ontology (Biological pathway; GO_BP) enrichment analysis for the 1000 most-divergent genes between control and homozygous iPSCs (based on FDR, Supplementary Data 2) using DAVID (Database for Annotation, Visualization and Integrated Discovery)[28] highlighted pathways and processes in which *SART3* has previously been implicated, such as cancer signalling[29] and stem cell pluripotency[30] (Fig. 6b). Analysis of only up- or downregulated DE genes found upregulated genes were enriched for terms such as regulation of transcription, development and axon guidance, whereas downregulated genes were enriched for metabolism terms, among others (Supplementary Data 2). Mass spectrometry (MS) analysis found 1002 proteins DE (adjusted *P*-value < 0.05) between control and homozygous cells (Supplementary Data 2). These were enriched for Gene Ontology terms related to mRNA splicing ($P = 0.01$, fold enrichment (FE) 2.38) and RNA ($P = 5.6 \times 10^{-4}$, FE = 10.5) and protein localisation to the Cajal bodies, a

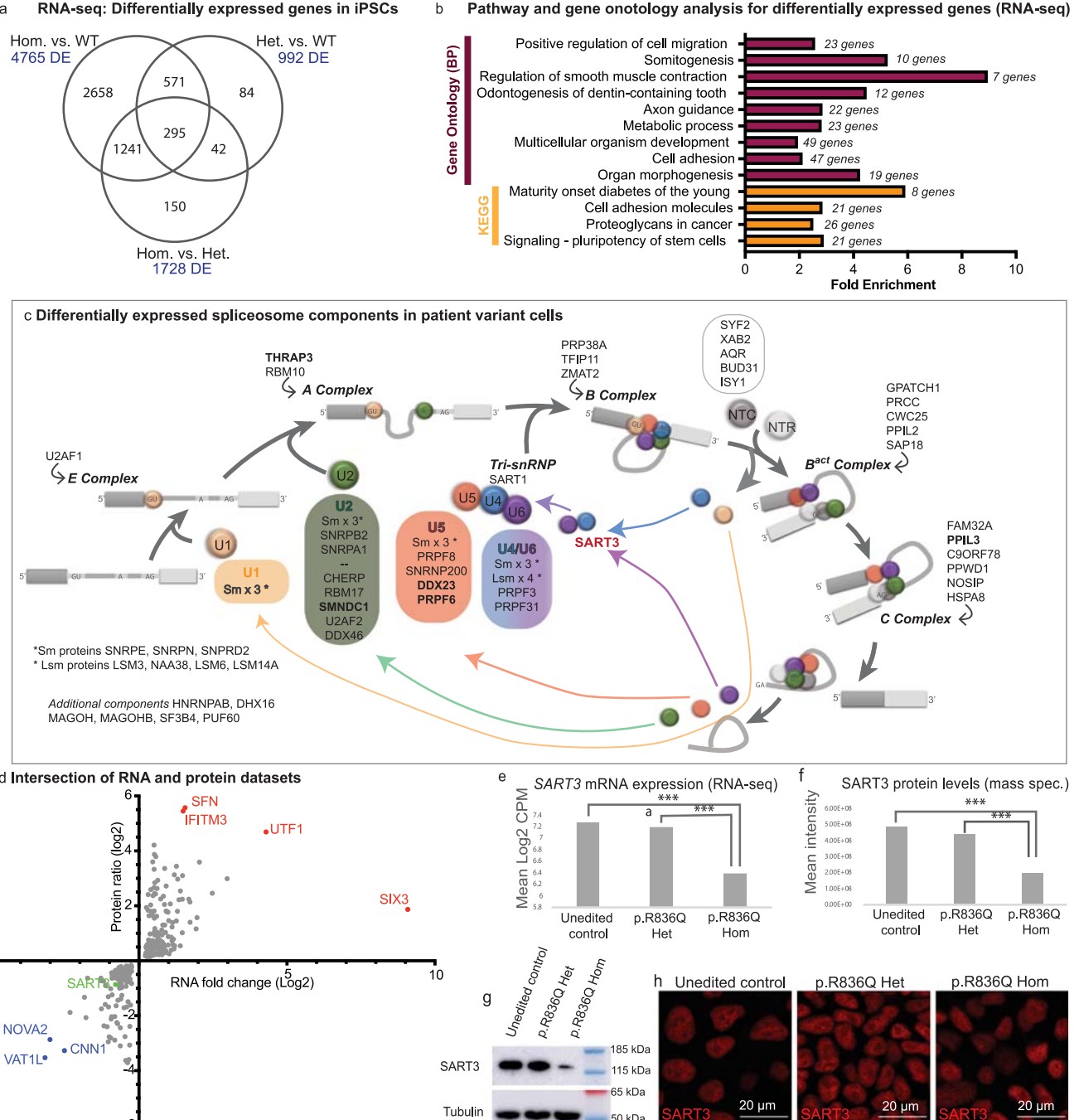

**Fig. 6 | The patient variant p.Arg836Gln disrupts numerous signalling pathways and leads to reduced *SART3* RNA and protein levels.** RNA-seq and MS analysis of heterozygous, homozygous p.Arg836Gln variant or non-edited control iPSCs. **a** A Venn diagram of differentially expressed (DE) genes between groups (FDR < 0.05). **b** Number of DE genes is shown in blue for each comparison KEGG and Gene Ontology (biological process) pathway analysis identified 13 pathways/gene ontologies with significant enrichment (*P* < 0.05 adjusted for multiple comparisons, Benjamini Hochberg FDR) represented in the topmost DE genes between control and homozygous iPSCs (1000 genes based on FDR). **c** A schematic of the major spliceosome. SART3 (red) has been implicated in recycling U4 and U6 snRNAs. Components that are DE in variant iPSCs are shown (RNA-seq or MS. Bold text = both with change in same direction). **d** Intersection of the DE genes (FDR < 0.05)

and proteins (FDR < 0.05) revealed a significant overlap (*P* < 10⁻⁹, one-sided Fischer's Exact Test) with 350 genes/proteins DE in the same direction in both datasets (plotted as log₂ ratio/fold change). Genes/proteins with the highest upregulation (red) or downregulation (blue) are shown. SART3 is downregulated in the homozygous variant cells (green). **e** *SART3* expression from RNA-seq (mean log₂ counts per million, CPM) *** = FDR = 1.52E-05 and 1.45E-04 respectively. Downregulation was confirmed by RT-qPCR – see Supplementary Fig. 6i. **f** MS found a significant difference in SART3 protein levels in homozygous iPSC line compared to control (****P* = 3.77E-05), or heterozygous line (****P* = 1.11E-04). **g** This was confirmed in western blot analysis – see also Supplementary Fig. 5p. **h** SART3 staining is nuclear in all three lines. RNA-seq and proteomic data analysis is provided in Supplementary Data 2.

nuclear structure important for snRNP maturation and where SART3 and SART3·U4/U6 snRNP complexes accumulate[31] ($P = 2.5 \times 10^{-9}$, FE = 15.4; Supplementary Data 2). SART3 plays a well-established role in spliceosome component recycling[15] (Fig. 6c) and spliceosome genes as a group were significantly enriched ($P < 0.003$) in DE genes in homozygous cells (31 genes DE, FDR < 0.05), where all but one are upregulated (median fold change = 1.3). These genes encode proteins associated with components from across the spliceosome (Fig. 6c), including U2, U5 and U4/U6 snRNAs, or associated B and C complexes (Fig. 6c). rMATS analysis of RNA sequencing data[32] did not find major differences in the number of splicing events such as exon skipping or intron retention between control and SART3 homozygous variant iPSCs, but differential transcript usage (DTU) analysis using equivalence classes[33] found 347 transcripts from 232 genes with DTU between control and homozygous cells (FDR < 0.05) (Supplementary Data 2). Confirmation of a subset of these in independent samples was carried out using RT-PCR (Supplementary 6k–m). Thus changes in gene expression and DTU are associated with the homozygous SART3 variant in the pluripotent state.

The RNA and proteomics data showed a statistically significant overlap ($P < 2 \times 10^{-16}$, Fischer's exact test) with 350 genes/proteins DE (FDR < 0.05) between homozygous variant and control cells in both datasets and a fold change in the same direction (Supplementary Data 2 and Fig. 6d). This includes SART3, with SART3 mRNA and protein levels significantly reduced in the homozygous p.Arg836Gln variant iPSC line compared to the heterozygous and unedited control (Fig. 6d–g FDR < 0.001 and g). This finding is unexpected as the missense variant is not predicted to cause nonsense-mediated decay and may indicate a disrupted auto-regulatory loop[34]. In iPSCs (Fig. 6h) or in testis organoids and neurons derived from these (Supplementary Fig. 5), the SART3.p.Arg836Gln protein shows a similar cellular localisation to WT. Other genes/proteins showing overlap in the two datasets include several that have a defined role in stem cell maintenance such as UTF1 (Undifferentiated Embryonic Cell Transcription Factor 1)[35], and SIX3 (SIX Homeobox 3) (Fig. 6d), both highly upregulated. Highly downregulated genes/proteins included NOVA2 ($\log_2$ fold change RNA = −3.01, protein = −2.87; Fig. 6d), which regulates alternative splicing in the brain, has a role in axonal pathfinding during cortical development[36], and underlies a severe neurodevelopmental disorder[37]. We decided to further explore NOVA2 interactions given its role in splicing in the brain and neurodevelopmental conditions.

## Potential interactions between SART3, NOVA2 and gonadal pathway genes

To further explore the potential interaction between SART3 and NOVA2 we utilised NTERA-2 cl.D1 (NT2/D1) cells, a multipotent cell line derived from human embryonic carcinoma cells that express SART3 and NOVA2 and differentiate into various cell types including neurons following appropriate stimulation[38]. We found that RNAi-mediated KD of SART3 correlated directly with a significant reduction in NOVA2 expression (Fig. 7a, b), confirming a genetic interaction of these two genes. NT2/D1 cells have also been employed as a model for testicular somatic cells as they express at least 40 testis-specific genes, and male pathways initiated by SRY, SOX9 and SF-1 are intact[39–41]. KD of SART3 resulted in no change in expression of gonadal/testis genes NR5A1 or SOX9 (Fig. 7e, f) but did result in significantly reduced FGF9 (Fig. 7c); a key component of testis differentiation[42]. Upregulation of the bi-potential gonad gene GATA4 was also observed in the cells with the highest KD (Fig. 7d). These findings provide a compelling link between disruptions to SART3 and the regulation of key gonadal genes that may underpin the testicular dysgenesis observed in our patients.

## Discussion

We describe here nine individuals with bi-allelic variants in SART3 and a syndrome characterised by 46,XY gonadal dysgenesis and neurodevelopmental defects. To our knowledge, this is the first report of bi-allelic SART3 variants causing a congenital syndrome.

Given this new role for SART3, we endeavoured to study its contribution to development using two animal models, Drosophila and mice. Our Drosophila studies demonstrated a critical and conserved requirement for SART3 in testicular and neuronal development. We were unable to further study this condition in mice as, despite multiple attempts, our CRISPR/Cas9 genome editing experiments failed to produce viable offspring carrying either the p.Arg836Gln or p.Glu211-Lys patient variant. This is somewhat surprising given that we successfully introduced the p.Arg836Gln change in iPSCs. Indeed, we attempted CRISPR/Cas9 mouse model generation when the tool was relatively new and guide RNA and Cas9 mRNAs were generated using an in vitro transcription method[43], which can introduce impurities causing embryo toxicity. More recently refined approaches such as ribonucleoprotein forms of CRISPR, which have been found to produce consistent results with higher editing efficiencies[44], may yield better success. Of note, we did recover mice with null alleles created by the gene editing procedure which were incompatible with viability. This lethality is consistent with previous studies in several organisms[15], and the absence of homozygous SART3 LOF variants in online databases such as GnomAD suggests this may also be the case in humans. It also suggests that our SART3 variants are hypomorphic alleles acting in an autosomal recessive mode. Indeed, heterozygous parents and all siblings studied are asymptomatic.

All children described here had severe neurodevelopmental symptoms from infancy including intellectual disability and global developmental delay. Neurological anomalies include agenesis/hypoplasia of the corpus callosum, enlarged ventricles and cerebellar atrophy. The overlapping findings are not stereotypical of other known conditions and there is no evidence of progressive neurodegeneration clinically. Our finding that SART3 may regulate the expression of the neurodevelopmental disease-causing gene NOVA2 highlights a potentially important genetic interaction that warrants further investigation.

Up to 20% of individuals with 46,XY DSD also report a condition of the central nervous system[45] and there is significant overlap between genes expressed in the testis and in the brain[46,47]. Yet gonadal phenotypes are uncommon in spliceosomopathies, with few reports and only fragmentary functional follow up[13,14]. Of the affected individuals we described here, those with a 46,XY sex chromosome complement had gonadal dysgenesis whilst 46,XX individuals appear to have unaffected ovarian development (confirmed for one pubertal girl and suspected in the others). This suggests that SART3 variants cause disruption specific to the early testis lineage. The degree of 46,XY gonadal dysgenesis varied between individuals, even in those carrying the same variant (ISR1 and ISR2), with some patients having evidence of early testis activity indicated by ambiguous or undervirilised male genitalia and underdeveloped or absent Müllerian structures, or in the case of one child, the presence of both Müllerian and Wolffian structures.

How SART3 variants affect testis development remains enigmatic, but our findings here may shed some light. Testicular differentiation is triggered by SRY expression in the pre-Sertoli cells of the bi-potential gonad at around 5–6 gestational weeks in humans[48,49]. This activates expression of SOX9 and other male-promoting factors including FGF9 and PGD2[50,51]. Sertoli cells then secrete AMH during weeks 8–9 of gestation, which is responsible for Müllerian duct regression in the male foetus. However by week 10 Müllerian ducts become insensitive to AMH[52]. Since SART3 shows strong speckled localisation to the nucleus in the Sertoli cells, and weaker expression in the cytoplasm of the human week 9 gestation testis, we postulate that SART3 may be especially required at this critical timepoint for testis development and subsequent Müllerian duct regression. Indeed testis-like organoids differentiated from iPSCs with homozygous SART3 variants were significantly smaller than controls and demonstrated widespread

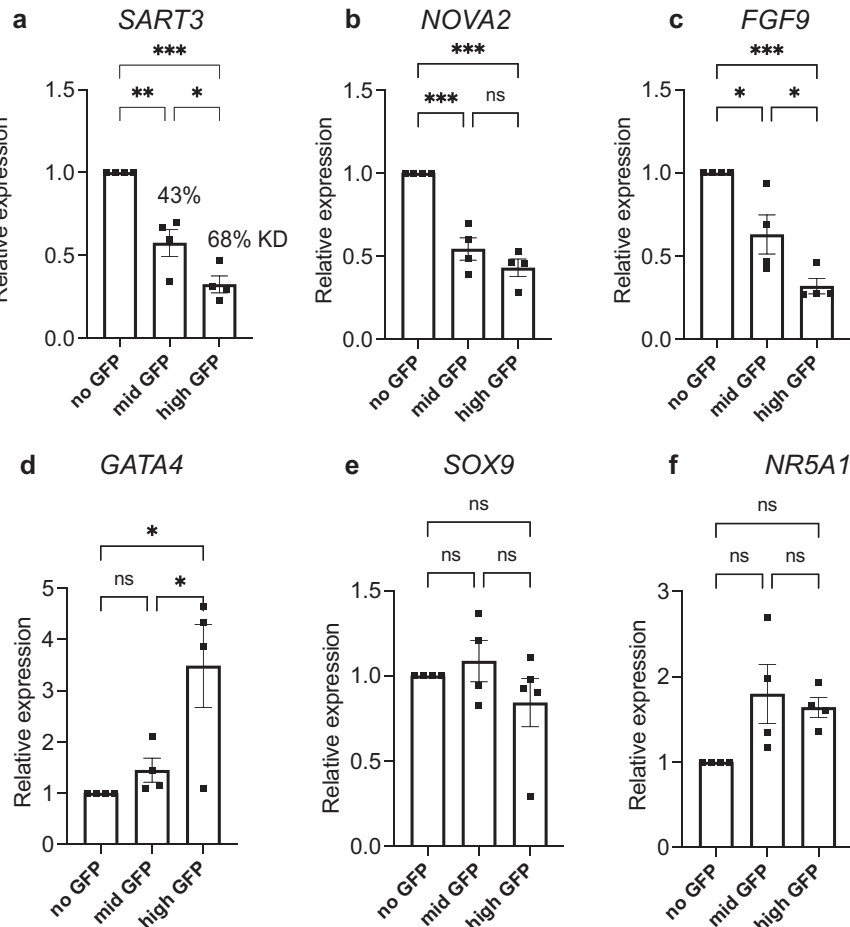

**Fig. 7 | Knockdown of SART3 in NT2/D1 cells highlights potential interaction between *SART3*, *NOVA2* and gonadal pathway genes. a**–**d** shRNA mediated KD of SART3 in GFP-sorted NT2/D1 cells. **a** Mid and high GFP cells had 43 and 68% KD of *SART3* mRNA respectively compared to non-GFP control cells, as determined by RT-qPCR. **b** *NOVA2* mRNA levels correlated with *SART3* KD. **c** *FGF9* levels were also reduced, and **d** *GATA4* expression was higher in KD cells. **e** *SOX9* and **f** *NR5A1* expression were not significantly different in KD cells. Expression is relative to the non-GFP control and shown as mean ± SEM. *n* = 4 independent experiments. A one-way ANOVA with Tukey's multiple comparison test was used. *P* values are *<0.05; **<0.01; ***<0.001.

apoptosis. Reduced expression of the intermediate mesoderm marker *PAX2*, the early gonadal gene *NR5A1* and the Sertoli cell factor *FGF9*, was observed. Changes in *FGF9* expression were also observed when *SART3* was KD in NT2/D1 cells. Mutations in *Fgf9* or its receptor *Fgfr2* result in sex reversal in XY mice[53–57]. In human foetal gonad cultures dysregulation of FGFR-mediated signalling affects testicular development[58], and some emerging evidence suggests that variants in the *FGF9* pathway may contribute to DSD, with mutations in *FGFR2* described in an XY individual with gonadal dysgenesis[59], and a gain of *FGF9* copy number in one *SRY*-negative 46,XX male patient[60]. Of note, in mice SOX9 regulates *Fgf9*, which forms a regulatory loop to maintain the expression of *Sox9*[61]. We did not observe a reduction in *SOX9* in our models, hence the specific mechanism by which SART3 might regulate *FGF9*, and the importance of this observation in the patient phenotype, remain unknown. Optimisation of our basic testis organoid model to include additional cell types and functional readouts may provide better insight into disease mechanisms in the future.

Whilst our stem cell-based studies indicated that *SART3* variants disrupt differentiation into both neural and gonadal lineages, the exact mechanisms underlying this disruption in patients remain to be discovered. The patient variants fall in two domains previously implicated in splicing - specifically in U6 snRNA binding[19] and U4/U6 snRNP recycling activity[19]. Transcriptomic analysis of iPSCs carrying a patient *SART3* variant did not show a significant increase in splicing events

indicative of a disruption to the spliceosome (i.e no increase in intron retention or exon skipping), although alternative transcript usage for more than 300 genes was observed. Variant *SART3* iPSCs also had widespread changes to gene expression, including upregulation of many spliceosome components. This may indicate a compensatory mechanism, as has been proposed in zebrafish *SART3* mutants (*Earl grey*). These fish had disrupted recycling of singular snRNPs back into U4/U6 di-snRNPs[22], and upregulation of numerous splicing-related genes (such as LSm and Sm factors)[22]. It was postulated that this extra synthesis of spliceosome components would allow increased de novo biosynthesis of tri-snRNPs, which would mitigate the spliceosome recycling defect. Indeed, comparison of the genes affected in the zebrafish mutant with DE genes from our homozygous variant iPSCs revealed 27 common genes, including LSm and Sm factors, suggesting the existence of a conserved compensatory mechanism.

Another potential explanation for the absence of major disruption to splicing in our iPSCs may be the cell type studied. Organ-specific vulnerability has been observed for many spliceosomopathies[13], and our patient phenotypes indicate that the CNS and testis are particularly sensitive to *SART3* variants. In vitro studies have suggested SART3 is needed as a splicing factor specifically in conditions where there is an increased requirement for the spliceosome[19]. Indeed, the brain and testis largely exceed the other tissues with respect to the expression of splice-variants[62–65] and greater expression of specific

signatures of splice factors suggests that alternative splicing plays a particularly crucial role in these tissues[66]. Thus, any tissue-specific vulnerabilities to *SART3* variants may not have been detected in our profiling of the iPSC pluripotent state. Nevertheless, our finding that a single nucleotide change in *SART3* caused significant shifts in the transcriptome and proteome as well as disrupted differentiation provides evidence supporting causation of this disorder.

Receiving a genetic diagnosis has significant benefits for individuals born with a DSD. It can put an end to the diagnostic odyssey and reduce unnecessary and often invasive testing. It can facilitate therapies and guide treatment to alleviate or prevent comorbidities, including monitoring the risk of gonadal cancer in patients with 46,XY gonadal dysgenesis. It also opens up avenues for further research into the disease process. Both DSD and neurodevelopmental disorders are underdiagnosed at the molecular level[67,68]. Our research indicates that the *SART3* gene should be investigated in babies born with 46,XY gonadal dysgenesis, ambiguous genitalia, or under-virilized male genitalia associated with gonadal dysgenesis, as it may provide a diagnosis of this syndrome before the appearance of additional comorbidities such as ataxia and intellectual disability. Heterozygous *SART3* variants have been implicated in disseminated superficial actinic porokeratosis (DSAP)[69,70] as well as various malignancies[15], with two of the variants we described reported as somatic mutations in COSMIC (p.Arg836Gln and p.Arg253*). On this basis, increased cancer monitoring in families with a *SART3* genetic diagnosis may be beneficial.

In summary, we present evidence that recessive variants in *SART3* cause a spliceosomopathy. We tentatively propose INDYGON syndrome (**I**ntellectual disability, **N**eurodevelopmental defects and **D**evelopmental delay with 46,X**Y GON**adal dysgenesis) to describe this condition. Our findings solidify an important link between variants in core spliceosome components and syndromic gonadal dysgenesis and highlight genetic interactions with key genes essential for testis and brain development. These findings will enable additional diagnoses and improved outcomes for patients born with these debilitating conditions.

## Methods

This research complies with all relevant ethical regulations and was approved by the following boards/committees: Human Research Ethics Committee, Royal Children's Hospital, Melbourne Australia (HREC22073). French Ethical Committee (2014/18NICB; registration no. IRB00003835), Bambino Gesù Children's Hospital (registration no. 1779_OPBG_2019), Assistance Publique Hôpitaux Marseille (reference PADS21-282). Written informed consent for participation in the study was obtained from all human research participants (or their guardians). All parents/guardians have seen and consented to the research and to publication of data within the context of the paper. Human foetal testis tissue was obtained following elective termination of pregnancy during the 1st trimester at the Departments of Gynaecology at Copenhagen University Hospital (Rigshospitalet) and Hvidovre Hospital, Denmark, following informed written and oral consent (ethics permit H-1-2012-007). None of the terminations were for foetal or pregnancy pathology. *Drosophila* work was carried out according to protocols approved by the University of Melbourne Institutional Biosafety Committee (IBC) reference no: 2017/023. Mouse work was conducted according to protocols approved by the University of Queensland Ethics Committee (IMB/435/13/NHMRC/ARC; IMB/445/12/NHMRC/BREED; IMB/232/13/NHMRC/ARC).

### Genetic testing
#### Families ISR2 and ISR2
**ISR1.** Causative variants in *CMA, MECP2, CDKL5, FOXG1, DAX1, L1CAM* and *ARX* and large genomic rearrangements had been ruled out prior to WES using Sanger sequencing and Copy Number Variant (CNV) arrays (Illumina Omni 2.5). ISR2: ISR2.2 CGH-array (Affymetrix

Cytogenetics Whole Genome 2.7M array chip and genome wide human SNP array 6.0) did not detect any causative genomic anomalies. CNVs in *ATRX* were also ruled out (Emory Genetics Laboratory, Decatur, GA, USA).

**ISR1 and ISR2.** Exome capture (Illumina TruSeq) and sequencing (Illumina HiSeq 2000) were performed by the Australian Genome Research Facility (AGRF, Melbourne, Australia). The 100 bp paired-end reads were aligned to UCSC hg19 using Novoalign version 2.07.17 (www.novocraft.com). Reads mapping to multiple locations were discarded; presumed PCR duplicates were discarded using MarkDuplicates from Picard (http://picard.sourceforge.net/). Variants were detected using the mpileup and bcftools view commands from SAMtools version 0.1.18[71,72] specifying parameters -C50 and -q13. Low-confidence variants were discarded using the vcfutils.pl varFilter script from the same programme. Variants were annotated against the UCSC KnownGene annotation, dbSNP132, 69 genomes from Complete Genomics, 1092 genomes from the 1000 Genomes Project (November 2010 and May 2011 releases) 3510 European ancestry exomes from 5400 American exomes sequenced as part of the NHLBI GO Exome Sequencing Project (ESP) using ANNOVAR 28 Nov 2011 version[73].

### Estimation of relatedness
The proportion of SNP alleles shared identically by descent (IBD) between the eight genotyped samples from ISR1 and ISR2 was estimated using the --genome analysis option of PLINK v1.07[16]. Parametric multipoint linkage analysis was performed using the programme MERLIN v1.1.2[74]. A fully penetrant autosomal dominant genetic model was applied, i.e. Pr(disease| (Aa or aa) = 1 and Pr(disease|AA) = 0, where a is the unknown disease-causing allele. The disease allele frequency was set to 0.00001. Allele frequencies obtained from the CEU HapMap population were used. MERLIN was used to calculate a heterogeneity LOD score (hLOD score). Haplotypes were produced using HaploPainter[75] using output files generated by the MERLIN linkage analysis.

Sanger sequencing was used to confirm *SART3* variant in ISR1 and ISR2 (primer details are provided in Supplementary Data 3). In all, 50–100 ng of genomic DNA was PCR amplified using Phusion DNA Polymerase with High Fidelity (HF) buffer (Finnzymes; F-530L) according to manufacturer's instructions – 98 °C for 30 sec, 35 cycles of 98 °C for 10 sec, 68 °C for 10 sec and 72 °C for 30 s, a final extension at 72 °C for 10 min. PCR products were purified using the MinElute PCR Purification Kit (QIAGEN; 28006) and sequenced at the AGRF. Sanger sequencing was used to screen 165 healthy women with of Moroccan Jewish descent for *SART3* variants (PCR amplification took place at Tel Aviv Sourasky Medical Center, Tel Aviv, Israel, with sequencing performed at AGRF, Melbourne).

**Family TUN1.** The parents and proband were sequenced using exon enrichment performed with Agilent SureSelect Human All Exon V4. Paired-end sequencing was performed on the Illumina HiSeq2000 platform with TruSeq v3 chemistry. Read files (fastq) were generated from the sequencing platform via the manufacturer's proprietary software. Reads were mapped with the Burrows-Wheeler Aligner, and local realignment of the mapped reads around potential insertion/deletion (indel) sites was carried out with GATK version 1.6. Duplicate reads were marked with Picard version 1.62 (http://broadinstitute.github.io/picard/). Additional BAM file manipulations were performed with SAMtools (0.1.18). SNP and indel variants were called with the GATK Unified Genotyper for each sample. SNP novelty was determined against dbSNP138. Novel variants were analysed by a range of web-based bioinformatics tools with the EnsEMBL SNP Effect Predictor (http://www.ensembl.org/homosapiens/userdata/uploadvariations). All variants were screened manually against the Human Gene Mutation Database Professional Biobase. The *SART3.*c.

NM_014706.3(SART3_v001):c.631G>A variant was confirmed by Sanger sequencing in all available family members.

**Family ISR3.** Trio exome sequencing was carried out at Rambam Medical Center on the Novaseq6000 platform (Illumina) using the IDT_xGen_Exome_Research_Panel_v2 kit (IDT). Mapping of the obtained reads to the reference genome (build GRCh37/hg19), variant calling, annotation and data analysis were using the Genoox data analysis platform Ltd (Genoox). Sequencing data was filtered on a trio-based paradigm to identify recessive, X-linked, and potential de novo variants in the proband. Variants were prioritised based on their effect on the protein and minor allele frequency <1% in general population databases, such as gnomAD and the Rambam Genetics Institute internal database of over 1500 Israeli exomes.

**Family ITA1.** Exome sequencing was performed on genomic DNA of the patient and parents using SureSelectQXT Clinical Research Exome V2 (Agilent Technologies) and run on a NextSeq500 sequencer (Illumina). Bioinformatic analysis was carried out by aligning sequences to the human reference genome (GRCh37) using BWA v0.7.5. Variants were called with GATK Unified Genotyper and annotated through Geneyx software (https://geneyx.com). Variants were evaluated in silico by using Deleterious Annotation of genetic variants using Neural Networks (DANN), Combined Annotation-Dependent Depletion (CADD), Polymorphism Phenotyping v2 (PolyPhen-2), Sorting Intolerant from Tolerant (SIFT) and Mutation Taster. Sequencing data was filtered on a trio-based paradigm to identify recessive, X-linked, and potential de novo variants in the proband. Variants were prioritised based on their effect on the protein and minor allele frequency <1% in general population databases, such as gnomAD, ALFA and TOPMED.

**Family FRA1.** WES was performed at AP-HP, Sorbonne Université, Trousseau Hospital, in the proband and both parents with the following: SeqCap EZ MedExome capture kit (Roche) and sequencing on Illumina NextSeq 500 with 151 bp paired-end reads. The BaseSpace cloud computing platform (with BWA 2.1 and GATK Unified Genotyper 1.6) and the VariantStudio v.3.0 software provided by Illumina were used for analysis. Sequencing data was filtered on a trio-based paradigm to identify recessive, X-linked, and potential de novo variants in the proband. DNA variants were prioritised according to the following criteria: high quality score (Q Phred score ≥30), minor allelic frequency (MAF) < 0.01 in GnomAD, conservation and predicted impact on coding and noncoding sequence using the classical in-silico tools (CADD, PolyPhen-2, SIFT, MaxEntScan).

## SART3 protein models
SART3 protein sequences were downloaded from NCBI and the ClustalX program v2.1 was used for sequence alignment. Crystal structures were visualised in the RCSB PDB 3D protein viewer (Mol*Plugin 3.29.0).

## SART3 plasmids and cell culture
The mammalian expression vector pCMV6-Entry-hSART3-FLAG (RC210837; OriGene) was used to create the variant SART3 expression vectors using site-directed mutagenesis. The QuikChange II XL Site-directed Mutagenesis Kit (Agilent Technologies) was used according to the manufacturer's instructions. Mutations were confirmed using Sanger sequencing and vector primers or primers within *SART3* coding region (Supplementary Data 3).

## SART3 variant immunofluorescence
HEK293t cells (ATCC CRL-3216) were maintained in D-MEM supplemented with 10% foetal bovine serum (FBS) and 2 g/L NaHCO$_3$. 5 h prior to transfection the cells were seeded at 80% confluence on to an eight-well chamber slide (Lab-tech). 100 ng of empty vector control, wild type or variant pCMV-SART3-FLAG expression vectors were transfected using Lipofectamine 2000 (0.5 μL/chamber, Invitrogen) with Gibco Opti-MEM - Reduced Serum Medium (Gibco, 31985-070) used to dilute the DNA and the Lipofectamine reagent before complexing. After 24 h cells were washed in phosphate buffered saline (PBS), fixed with 4% paraformaldehyde (PFA), permeabilised and blocked with blocking buffer 5% bovine serum albumin (BSA), 10% donkey serum in 0.1% Triton-X in PBS (PBTX). Cells were incubated overnight with antibodies in 1% BSA, 2% serum and 0.1% PBTX. Cells were washed three times with PBS and incubated in secondary antibodies. DAPI was used for nuclear counterstaining, and secondary antibody only staining was used to control for unspecific binding. Cells were imaged using the Zeiss LSM780 confocal microscope. Two independent experiments yielded the same results. All antibody details are provided in Supplementary Data 4.

## RNA extraction, cDNA synthesis, quantitative PCR
RNA isolation was performed using the RepliaPrep RNA miniprep system (Promega, Z6011) following the manufacturer's instructions for $1 \times 10^2$ to $5 \times 10^5$ adherent cells. RNA integrity and concentration were measured by UV spectrophotometry (NanoDrop ND-1000, Thermo Fisher Scientific). Reverse transcription was carried out using the GoScript Reverse Transcriptase system (Promega, A5001) with random primers. RT negatives were performed for each sample at half reaction or for pooled samples from replicates. RT-qPCR (GoTaq qPCR master mix, Promega, A6002) was carried out with primers (Supplementary Data 3). qPCR data were collected on the Roche Lightcycler 480 II system, using the LC480 analysis software version 1.5.1.62. ΔΔCT values were calculated for each target and expressed as a ratio of the control (empty vector) for NT2/D1 experiments, or as a ratio of day 0 control iPSC line. Graphpad Prism v9 was used for statistics and to plot graphs.

## Western blot analysis
**Transient expression experiments.** HEK293t cells were cultured as above and were seeded at approximately 80% confluency on a 24-well plate ($3.2 \times 10^5$ cells). In all, 0.4 μg wild type or variant pCMV-SART3-FLAG or an empty vector control were transfected using Lipofectamine 2000 (0.5 μL/chamber, Invitrogen, 11668019) as described above. After 24 h cells were washed in PBS and lysed using NET lysis buffer (50 mM Tris HCl pH7.4, 250 mM NaCl, 0.5% NP-40, 5 mM EDTA and protease inhibitors (cOmplete ULTRA tablets, Roche, 05892970001)). Total protein was assessed using a Pierce BCA protein assay (Invitrogen, 23227), and 1–3 μg protein was run on a NuPAGE™ 10%, Bis-Tris, 1.0 mm, 12-well gel (Invitrogen, NP0302) with MOPS SDS running buffer (Invitrogen, NP0001), transferred to PVDF membrane, blocked using 5% skim milk powder/Tris Buffered Saline with 0.1% Tween (TBST) and incubated with primary antibody overnight at 4 °C. After washing, the membrane was incubated with swine anti-rabbit HRP or rabbit anti-mouse HRP at room temperature for 2 h. After blot washing the Amersham ECL Prime western blotting detection reagent (Cytiva, RPN2232) was used and visualised with the GE Amersham Imager 680 and integrated software v2.0.0 was used to collect data. Blots were washed and then incubated with anti-Beta Tubulin loading control (HRP) (1:10000, Abcam, ab21058) and detected as above.

**Confirmation of MS results.** MS results were validated using western blot analysis (as described above) for a subset of proteins (Supplementary Fig. 6). Total protein was extracted from three independent iPSC pellets with RIPA buffer (50 mM Tris-HCl, ph7.6; 150 mM NaCl; 1% NP-40; 0.5% sodium deoxycholate; 0.1% SDS with protease inhibitor cocktail (cOmplete ULTRA tablets, Roche, 05892970001)), 200–300 μL per 1–2 million cells. 5ug of total protein was run per lane. All antibody details are provided in Supplementary Data 4. Uncropped and unprocessed western blot images can be found in the Source Data file.

## SART3 staining in human tissues

Human foetal tissues (week 9) were dissected in ice-cold PBS and the isolated testis were immediately fixed in formalin, embedded in paraffin. Sectioned tissue underwent de-wax treatment consisted of two 2 min washes in Xylene, two 2 min washes in 100% ethanol, 1 min in 90% ethanol, 1 min in 80% ethanol, 1 min in 70% ethanol, 1 min in 50% ethanol and one dip in distilled water, slides were then stored in 1x PBS. Antigen retrieval consisted of two 5 min washes in 0.1 M citrate buffer (2.94 g trisodium dehydrate in 900 mL distilled water, at pH 7) in the microwave on high, then cooled to room temperature. Slides were blocked with horse serum (10%) in humidified incubation chamber at room temperature for 2 h; primary antibody incubations were performed overnight at 4 °C. Secondary antibody reactions were performed in a dark humidified chamber at room temperature for 2 h. Slides were washed in PBS for 5 min in a black box at room temperature. Slides were mounted with Fluorsave (Millipore, 345789) containing the nuclear stain DAPI. Imaging was performed on a confocal microscope (LSM780, Zeiss). All antibody details are provided in Supplementary Data 4.

## *Drosophila* studies

**Fly stocks and husbandry.** The fly stocks used in this study were *w*[1118], *tub*-Gal80ts; *tub*-Gal4/Tm6b, *elav*-Gal4; UAS-CD8::GFP, *repo*-Gal4 (UAS-GFP), *traffic-jam*-Gal4/SM6B; UAS-CD8::GFP, *tj*-Gal4 Don Juan-GFP. Unless otherwise stated, the lines were obtained from Bloomington Stock Center. The RNAi lines used were RNAi-B BL58168 (Bloomington) and RNAi-V V107063 (Vienna stock centre). All flies were raised on standard molasses-based food at 25 °C.

**Crosses: embryonic knockdown.** For global expression, *tub*Gal80ts; *tub*Gal4/TM6B was crossed with RNAi lines or *w*[1118] at the permissive temperature of 29 °C. Crosses at the non-permissive temperature of 26 °C were also analysed as controls. RNAi lines or *w*[1118] were crossed with *elav*-Gal4, UAS-CD8::GFP/CyO at 26 °C and 29 °C for neuronal expression or with *repo*-Gal4, UAS-GFP/TM6B at 26 °C and 29 °C for glial expression. Egg lays were performed on apple juice agar plates. Embryos were dechorionated in 50% bleach for 3 min, rinsed in water, then transferred to a 1:1 mix of Heptane and 4% paraformaldehyde and placed on an orbital shaker for 20 min. The solution was then replaced with a 1:1 mix of heptane and methanol and embryos were vortexed for 2 min. Liquid was then replaced with 100% methanol and embryos were stored at −20 °C. Immunofluorescence: Embryos were rehydrated with 3x washes with PBT (PBS 0.2% Triton-X), then blocked with PBT + 5% horse serum + 0.1% BSA for at least an hour at room temperature. Incubation with primary antibodies was performed overnight at 4 °C in PBT + 2.5% horse serum + 0.05% BSA. Embryos were rinsed 3x in PBT and then incubated with secondary antibodies overnight at 4 °C. After 1x PBT washes with DAPI added 1 mL in 10 mL, 2x PBT washes, embryos were incubated with a glycerol/PBS series (50%, 70%, 90%) allowing the embryos to sink in each dilution. Embryos were mounted on a slide in-between two coverslips attached with glue to allow mobility. Imaging was performed on the Zeiss LSM 780. Embryo images are a maximum projection of a stack of 5–10 images encompassing the whole embryo.

**Adult fertility and gonadal analysis.** For global expression, *tub*-Gal80ts; *tub*Gal4/Tm6B was crossed with RNAi lines or *w*[1118] at the permissive temperature of 29 °C, or 26 °C as a control. Fertility was tested by crossing single males with virgin *w*[1118] females. For gonadal somatic cell expression, RNAi lines or the *w*[1118] control was crossed with *tj*-Gal4/SM6B; UAS-CD8::GFP at 18 °C and offspring were shifted to 29 °C at early pupal stages (due to observed lethality when crosses were performed at 29 °C). For Don Juan-GFP experiments, *tj*-Gal4; Don Juan-GFP adults were crossed to *w*[1118] and RNAi lines. Testes were analysed from males aged 14 days. Testis and ovaries were dissected in

PBS containing 0.1% Triton X-100 (PBT) and fixed in 4% formaldehyde for 30 min. After rinsing (in PBT), testes were blocked in PBT with 5% goat serum (Sigma) for 1–2 h at room temperature, and then incubated with primary antibody in the same solution overnight at 4 °C. After rinsing in PBT, testes were incubated for 1–2 h in PBT with 5% serum and secondary antibodies at room temperature. Following rinsing, testes were mounted on glass slides in Vectashield (VectorLaboratories) for confocal analysis. Imaging was performed on the Zeiss LSM 780 or Zeiss LSM 800. All antibody details are provided in Supplementary Data 4.

## Quantification of Rnp4f knockdown in *Drosophila*

RNAi lines were crossed with the *tub*Gal80ts; *tub*Gal4/Tm6B line, and shifted to the permissive temperature (29 °C) at day 1 post eclosion. Flies were collected after 7 days. For each group (control or KD flies) 6x adult flies were collected and RNA prepared using RNeasy Mini kit (Qiagen). RNA quantification was determined using the Qubit 4 Fluorometer and RNA quality assessed on an Agilent 2200 TapeStation (all RIN values were above 9.8). cDNA synthesis was carried out using SensiFAST cDNA Synthesis kit (Bioline). Droplet Digital PCR (ddPCR was performed using predesigned Taqman gene expression assays for Rnp-4f (Dm01799055_g1, Thermo Fisher Scientific) or Rpl32 (Dm02151827_g1, Thermo Fisher Scientific). Expression of *Rnp4f* was calculated as a percentage of the control in each RNAi line KD.

## Creation of iPSCs carrying SART3 p.Arg836Gln patient variant

iPSCs carrying the *SART3* NM_014706.3: c.2507G > A. p.Arg836Gln variant in homozygous or heterozygous form or unedited controls were derived from human foreskin fibroblasts (American Type Culture Collection number PCS-201–010) at the MCRI Gene Editing Facility using a protocol that combines reprogramming and gene editing in one step[76]. Briefly, episomal reprogramming plasmids (pEP4E02SET2K, pEP4E02SEN2L, pEP4E02SEM2K, and pSimple-miR302/367), mRNA encoding SpCas9-Gem, a plasmid encoding a single guide RNA (TGGCCCGACCGTTTGGTTT) that cleaves 1 bp away from the intended c.2507 G > A conversion, and a repair template plasmid comprising ~0.6 kb homology arms flanking the introduced mutation {chr12:108524972-108525968, UCSC Genome browser hg38} corresponding to sequence with *SART3*, cloned into the pSMART-HCKan plasmid vector were used. The repair template included a 3 bp synonymous change to act as a Cas9-blocking mutation and to facilitate the identification of gene-corrected iPSC clones by allele-specific PCR. $1 \times 10^6$ PCS-201-010 fibroblasts were electroporated (1400 V, 20 ms, 2 pulses) using the Neon transfection system (Thermo Fisher Scientific) and plated over 4 wells of a Matrigel-coated (BD Biosciences) six-well dish in fibroblast medium (DMEM + 15% FBS). Medium was switched to Essential 8 medium (E8 medium without TGFb, Gibco) supplemented with 100 μM sodium butyrate for 3 days and changed every other day until first iPSC colonies appeared (~2 weeks). Media was then switched to E8 and individual iPSC colonies were picked and expanded. Using this method (simultaneous reprogramming and gene-editing) all iPSC colonies picked are clonal and therefore don't require single cell cloning. 48 colonies were picked and screened. To identify correctly targeted iPSCs, genomic DNA was isolated using the DNeasy Blood & Tissue Kit (QIAGEN) and PCR analysis was performed using primers that flank the oligodeoxynucleotides (ODNs) flanking the 5′ and 3′ recombination junction. Heterozygous and homozygous clones were distinguished using ODNs that flank the intended target site (Supplementary Data 3). Sanger sequencing using flanking primers confirmed gene-correction, clonality and/or absence of indel mutations. Loss of reprogramming vectors was confirmed by PCR and karyotype was confirmed by Infinium CoreExome-24 DNA microarray (Illumina); cells tested negative for mycoplasma contamination. Pluripotency of iPSC lines was confirmed by flow cytometry,

immunofluorescent staining and creation of embryoid bodies (Supplementary Fig. 4).

## Proliferation assays in iPSCs

Proliferation assays were carried out using the Click-iT™ EdU Cell Proliferation Kit for Imaging (Invitrogen). Briefly, iPSCs were plated on eight-well chamber slides coated with Vitronectin (Stem Cell Technologies), and grown in E8 medium (Thermo Fisher Scientific) supplemented with 100 μM sodium butyrate. After 24 h, half of the media was replaced with E8 + 20 μM EdU (final conc. 10 μM). Cells were then incubated for 40 min, washed, fixed and permeabilised as per manufacturer's instructions. 200 μL of Click-IT reaction cocktail was added per chamber, incubated for 30 min and then washed. Hoechst 33342 staining was then performed according to kit instructions. Slides were imaged on the Zeiss LSM 780 confocal. Total cell numbers (Hoechst) and EdU positive cells were calculated using Image J (Version 2.9.0/1.53t) for five wells per cell genotype.

## Embryoid body formation

iPSCs were seeded in ultra-low adherence 96-well plates and cultured in E8 media (Stemcell Technologies) with 0.5% polyvinyl alcohol (Sigma) for 24 h. Subsequently, the cells were cultured for 2 weeks in E8 media, refreshed every 2–3 days, then plated onto vitronectin-coated 8 well chamber slides and cultured in E8 medium for 3 weeks. EBs were fixed with 4% Paraformaldehyde for 10 min, permeabilised in 0.2% Triton X-100 (Sigma) for 10 min and blocked in 2% Bovine Serum Albumin (Life Technologies) for 60 min at room temperature. Cells were then incubated with primary antibodies at 4 °C overnight, followed by an incubation with secondary antibodies for 60 min at room temperature. The coverslips were mounted on slides with DAPI to stain the nuclei (VectorLabs). Images were captured with an LSM 780 confocal microscope running Zen Black software (Zeiss version 2.3 SP1). Antibodies used are detailed in Supplementary Data 4. Images for SMA and SOX17 are a single slice. MAP2 are maximum projections of a stack of 10 slices at 0.8 μm intervals to allow visualisation of axonal projections.

## Neuronal differentiation

**Lentivirus production.** Lentivirus carrying NGN2 and the reverse tetracycline transactivator (rtTA) gene was prepared by first plating $4 \times 10^6$ HEK293t cells in a T-75 and grown in 5% FCS in DMEM/F12 (Gibco). The following day cells were transfected using Lipofectamine 2000 (Thermo Fisher Scientific) with plasmid DNA from either pTet-O-Ngn2-puro (52047, Addgene) or FUW-M2rtTA (20342, Addgene) and pMDLg/pRRE (12251, Addgene), pCMV-VSV-G (8454, Addgene), p-RSV-REV (12253, Addgene), at a DNA molarity ratio of 4:2:1:1, respectively. Viral supernatant was collected at 24, 48 and 72 h, filtered through a 0.45 μm membrane (Millipore) and concentrated by centrifugation at 85,000×g for 2 h at 4 °C in a Sorvall WX 100 Ultra Ultracentrifuge (Thermo Fisher Scientific). The supernatant was discarded and viral pellet resuspended in PBS (Thermo Fisher Scientific). Typically, 0.25–1 μL of concentrated virus was used for each neural differentiation.

## NGN2-based neural differentiation

For this study cortical excitatory neurons were generated by the expression of NGN2 in iPSCs. iPSCs cells were plated at 25,000 cells/cm² in a 24-well plate coated with 15 μg/mL Laminin (Sigma). The following day, cells were transduced with NGN2 and rtTA lentivirus. NGN2 gene expression was activated by the addition of 1 μg/mL doxycycline (Sigma), referred to as differentiation day 0. Cells were cultured in neural media consisting of 1:1 ratio of DMEM/F12: Neurobasal media supplemented with (all reagents from Thermo Fisher Scientific B27 (17504-044), N2 (17502-048), Glutamax (35050-060), NEAA (11140-050), β-mercaptoethanol, ITS-A (51300-044) and penicillin/streptomycin (#15140-122). and

1 μg/mL puromycin (Sigma-Aldrich was added for three days at which point neurons were supplemented with 10 ng/mL BDNF (Peprotech) and lifted with Accutase (STEMCELL Technologies) to chamber slides or plated at $1 \times 10^5$ in 24-wells for gene expression studies. To inhibit the overgrowth of proliferating cells, 2.5 μM Ara-C hydrochloride (Sigma) was added at day 7 for 48 h. Three independent NGN2 differentiation experiments were carried out. Staining was carried out as per HEK293t cells (detailed above). Imaging was performed on the Zeiss LSM 780 using the Zen Black software (Zeiss version 2.3 SP1). Images are a maximum projection of a stack to allow visualisation of axons. All antibody details are provided in Supplementary Data 4.

## Gonadal organoid differentiation

The three iPSC lines (PCS_201_010 control, heterozygous SART3 p.Arg836Gln variant or homozygous SART3 p.Arg836Gln variant) were expanded in E8 (Thermo Fisher Scientific). One day prior to commencement of the differentiation, iPSCs were plated at 10,000 cells/cm² on a 24-well plate coated with Vitronectin (Stem Cell Technologies, Vancouver, Canada), with the addition of RevitaCell (Gibco, Thermo Fisher Scientific) in a final concentration of 1:100. The following day (differentiation day 0; D0) media was replaced with Essential 6 (E6): Essential 4 media (Thermo Fisher Scientific) supplemented with 500 μL Holo-Transferrin (Sigma-Aldrich, St. Louis, MO) and 1 mL Insulin (Sigma-Aldrich). For the first four days the culture medium was supplemented with 3 μM CHIR99021 (R&D Systems, Minneapolis, MN) (D0-D4, inclusive). From day 4 until day 7, the culture medium was supplemented with 200 ng/mL FGF9 (R&D Systems), 1 μg/mL Heparin (Sigma-Aldrich) and 10 ng/mL BMP4 (R&D Systems). Organoids were generated on day 7. The monolayers were dissociated using TrypLE Select (Gibco), the reaction was neutralised with DMEM (High Glucose DMEM with 10% FCS and 1% Glutamax). For each organoid, 350,000 cells were aliquoted into 1.5 mL centrifuge tubes, the tubes were centrifuged three times at 400×g for 3 min, rotating the tubes 180 degree between each centrifugation. Using a wide bore tip, the pelleted cells were transferred on to transwell filters (Corning) in a six-well plate. The organoids were cultured in E6 without any additional growth factors until day 10. From day 10 until day 21, the organoids were treated with 500 ng/mL PGD2 (Cayman Chemicals). Throughout differentiation, media was changed every 2 days.

## Gonadal organoid immunofluorescence

Day 14 and day 21 organoids were fixed with 4% PFA/PBS for 5 min at room temperature then washed in PBS three times for 5 min each. Blocking buffer was added (0.1% PBTX + 5% BSA + 10% donkey serum) for at least 2 h at room temperature. Primary antibodies were added in diluent (1:4 Blocking buffer:PBTX) and left to incubate for 48 h at 4 °C. The organoids were washed three times with PBS for 5 min each. Secondary antibodies were incubated for at least 4 h at room temperature, or overnight at 4 °C. Antibody details are provided in Supplementary Data 4. The organoids were washed in PBS for 5 min then DAPI added (5 μL in 200 mL PBS), followed by a final PBS wash. Imaging was performed on either the Invitrogen EVOS M5000 (low magnification) or Zeiss LSM 780 using the Zen Black software (Zeiss version 2.3 SP1). Secondary-only stains were used to control for non-specific binding.

## RNA sequencing

The three iPSC lines (PCS-201-010 unedited control; PCS-201-010 heterozygous SART3 p.Arg836Gln variant or PCS-201-010 homozygous SART3 p.Arg836Gln variant) were expanded from frozen stocks three independent times to create triplicate cell pools for RNA sequencing. RNA was extracted using the ReliaPrep RNA Miniprep System (Promega, #Z6011), following manufacturer's instructions. RNA sequencing was carried out at the Victorian Clinical Genetics Services (VCGS), with Illumina stranded mRNA library prep and

sequencing of paired 150 base-pair paired end reads and 30 million reads per sample on the Illumina Novaseq6000. RNA sequencing reads were aligned to the hg38 version of the human reference genome using STAR (version 2.5.2)[77] in two-pass mode. Duplicated read pairs were then removed using PicardTools, as we identified at least one sample with a high level of duplicates (>90% of reads). Aligned reads were summarised to gene-level counts using featureCounts[78] (version 1.5.0) and the Gencode version 20 gene annotation. Gene-level counts were analysed using edgeR v3.34.1[79] with separate design matrix factors for controls, heterozygous and homozygous. Differential expression of spliceosomes genes as a group was performed using camera[80] and ROAST[81]. Differential transcript analysis was performed using the methods similar to those in[33] where equivalence class counts (ECCs) were generated by salmon[82] and differential ECCs was tested for using satuRn v1.0.0[83]. The Integrated Genomics Viewer (IGV) v2.12.3 was used to visualise reads.

The Database for Annotation, Visualisation and integrated Discovery (DAVID; 2021 update) was used for KEGG pathway analysis or Gene Ontology (biological pathway) enrichment analysis. Adjustments for multiple testing using the Benjamini method were carried out and KEGG pathways with an adjusted $P$-value < 0.05 were considered. RNA-seq results were validated using RT-qPCR for a subset of genes (Supplementary Fig. 6). Splicing changes in the homozygous group compared to the wild type group were examined using rMATs version v4.1.2. rMATS was run with the deduplicated aligned reads and version 40 of the Gencode human annotation.

## Confirmation of differential transcript usage identified by RNA-Seq

RNA was isolated from iPSCs using the ReliaPrep RNA cell Miniprep kit (Promega), quantification was performed using NanoDrop™ One Spectrophotometer (Thermo Fisher Scientific). 1 µg of total RNA was reverse transcribed into cDNA using GoScript™ Reverse Transcriptase with random primers (Promega), according to manufacturer's instructions, the cDNA was diluted to 5 ng/µL. PCR was performed using Phusion HighFidelity GC Mastermix kit. After optimisation, cycling conditions were as follows, 30 cycles (THYN1 and DRAM2) 32 cycles (SLC3A2), primer dependent annealing temperature, annealing time: 10 s, extension: 15 s) using the following primer pairs THYN1.ex5.RT.F and THYN1.ex7.RT.R';DRAM2.ex1.RT.F and DRAM2.ex4.RT.R (annealing temp 63 °C); SLC3A2.ex1.F and SLC3A2 ex.3.R (annealing temperature: 67 °C). The PCR products obtained were analysed on a TapeStation 2200 Instrument using D1000 screen-tapes (Agilent Technology). Statistics were performed on the calibrated concentration readings for each amplicon using Graphpad Prism v9. Primer sequences are in Supplementary Data 3.

## Liquid chromatography–mass spectrometry

Cell pellets were processed using sodium deoxycholate (SDC) solubilisation essentially as described previously[84]. Using a Dionex UltiMate 3000 RSLCnano system equipped with an Acclaim PepMap RSLC analytical column (75 µm x 50 cm, nanoViper, C18, 2 µm, 100 Å; Thermo Fisher Scientific) and an Acclaim PepMap 100 trap column (100 µm x 2 cm, nanoViper, C18, 5 µm, 100 Å; Thermo Fisher Scientific), the tryptic peptides were analysed on a QExactive Plus mass spectrometer (Thermo Fisher Scientific) operated in data-dependent acquisition (DDA) mode. The instrument was set to automatically switch between full scan MS and MS/MS acquisition. Each survey full scan (m/z 375–1575) was acquired in the Orbitrap with 70,000 resolution (at m/z 200) after accumulation of ions to a $3 \times 10^6$ target value with maximum injection time of 54 ms. Dynamic exclusion was set to 15 s. The 12 most intense multiply charged ions ($z \geq 2$) were sequentially isolated and fragmented in the collision cell by higher-energy collisional dissociation (HCD) with a fixed injection time of 54 ms, 17,500 resolution and automatic gain control (AGC) target of $2 \times 10^5$. MS raw files were analysed with MaxQuant v1.6.5.0 to obtain protein identifications and their respective label-free quantification values using in-house standard parameters. Downstream statistical analysis was performed in Perseus v1.6.2.3[85] and LFQ-Analyst v1.2.3[86].

## SART3 knockdown in NT2/D1 cells

A short hairpin RNA (shRNA) against *SART3* cloned into a RNAi-Ready pSIREN-RetroQ-ZsGreen vector (Clonetech) was transfected into NTERA-2 cl.D1 (NT2/D1) cells (American Type Culture Collection, # CRL-1973). Cells were cultured in DMEM + 10 % FBS. NT2/D1 cells were seeded on a six-well plate at 60% confluency. Transfection with the *SART3* knockdown vector took place after 24 h, when cells had reached 80% confluency. Each well was transfected with 5000 ng of the shRNA plasmid diluted in 250 mL of Opti-MEM medium and with 17.5 mL of Lipofectamine 2000 (Invitrogen) diluted in 250 mL of Opti-MEM medium. Cells were harvested for sorting after 48 h, with 0.5 mL of 0.025% trypsin applied to each well. Cells were then passed through a cell strainer (BD Falcon, 352235) to ensure single cells. Flow cytometry was carried out on the inFlux v7 Sorter, and cells were separated into non-transfected cells (no GFP), medium GFP, and high GFP. RNA was extracted and cDNA made as above. RT-qPCR was carried out for *SART3, NR5A1, SOX9, FGF9, WT1, NOVA2, GATA4* as described above with *GAPDH* serving as a housekeeping gene. The results were analysed in accordance with the ΔΔCT methodology, with expression levels normalised to that of the control cells.

## Genome editing in mice

We attempted disease modelling of *SART3* patient variants in mice using CRISPR/Cas9-mediated genome editing.

## ISR1/2 SART3 variant NM_014706:exon17:c. 2507G > A:p.R836Q (Exon 17)

In order to convert Arg to Gln in the mouse sequence it was necessary to change 2 bases of a codon. We attempted to introduce this variant at two facilities using the strategies outline below.

**University of Queensland, Australia.** Optimized CRISPR Design (http://crispt.mit.edu) was used to identify CRISPR/Cas9 guide RNA targets. For each target, upper and lower oligo pairs were annealed and inserted into PX330 to generate sgRNA template for IVT[87]. All sgRNAs were transcribed using MEGAshortscript T7 Kit (Ambion) and Cas9 nickase (Cas9n) or Cas9 mRNA IVT was performed using mMESSAGE mMACHINE T7 Transcription Kit (Ambion) from linearised pBS-Cas9n or pBS-Cas9. IVT sgRNA and Cas9n or Cas9 mRNA were purified with MEGAclear Transcription Clean-Up Kit (Ambion). Repair ssODNs (IDT) were designed with mutations to model patient variants incorporated, as well as mutations in the protospacer adjacent motif (PAM) recognition site to prevent re-targeting. Injection cocktails consisting of 10–20 ng/µL Cas9 mRNA, 5–10 ng/µL sgRNA and 10–30 ng/µL ssODN repair template (as detailed in Supplementary Table 3) were micro-injected into the male pronucleus of one cell embryos derived from superovulated C57BL/6J females. Injected embryos were cultured overnight and the next day 2-cell stage embryos were transferred into pseudo-pregnant CD1 females using standard methods. For most exon 17 experiments at UQ, a Cas9 nickase strategy was used to minimise off-target effects[88]. Guide RNAs were designed to target the exon 17 sequences 5'AGGCTGGTCACTAACAGGGC3' (upper strand) and 5'GAGGTCCTTGACGGTGCCGT3' (lower strand), arranged with PAM sequences out. Cas9 nickase mRNA was used and the ssODN repair template 5'TGGCCTGCCCTTTTCCTGCACCAAAGAGGAGCTCGAGGA-CATTTGTAAGGCTCACGGCACCGTCAAGGACCTCAGGCTGGTCAC-TAAC\CAG\GCTGGCAAGCCGAAGGTGAGTGGGGATGGTGGGCTTG GGTCGTCTGAGCTGGATACACCTTTCAAGCTCTGACTCCACATTG GCGAGT 3' was used to introduce the R836Q mutation. On one occasion, a truncated 18-mer gRNA was tested (5'CCGTCAAG

GACCTCAGGC3')[89] and Cas9 mRNA was used, with the same ssODN as above. Primers to detect mutations were Exon17F 5'CCCTGGAGAAA-CACAAACTC3'and Exon17R 5'TGGCTACAGACTTACCCCTC3' (374 nt product), Exon15F with Exon17Rmutate 5'CCACTCACCTTCGGC TTGCCAGCCTG3' (142 bp product) and Exon17R with Exon17Fmutate 5'AGGACCTCAGGCTGGTCACTAACCA3' (281 bp product).

## Walter and Eliza Hall Institute, Australia

A sgRNA with sequence ACCTCAGGCTGGTCACTAAC was used to create double stranded breaks within the Sart3 locus. ssODN: GAGCTCGAGGACATTTGTAAGGCCCACGGCACCGTCAAGGACCTCAG GCTGGTCACTAACCAGGCTGGCAAGCCGAAGG TGAGTGGGGATGGT GGGCTTGGGTCTGTCTGAGCTGGATACACC was used to introduce the p.R836Q mutation. Genotyping: forward primer 5'ATCTCTGG CCTGCCCTTTTC3' and reverse primer 5'CGTCTGACTGGCTGTCAC TG3'. Cas9 protein, sgRNA and ssOligo were microinjected into single cell zygotes from C57BL/6 mice.

## TUN1 SART3 variant NM_014706:exon4:c. 631G > A:p.E211K (TUN1)

**University of Queensland, Australia**. The corresponding mouse mutation is Sart3 NM_016926:exon4:c.G634A:p.E212K. For this variant two targeting strategies were employed. The first one (#1) used crRNA ccaggcctttcgtcatgtgcagg as a guide and tttaggtccaaacatttggctagagtat ggccagtactcagttggtggcattggtcagaaaggtggccttgagaaggttcgctctgtctttAa aagagccctgtcctctgttggTctgcacatgacgaaaggcctggccatctgggaggcctac cgagagtttgaaagcgccatcgtggaggctgctcgggtgagtccag as a repair template. This would produce the KI mutation 29 bp upstream of the expected cutting site and included a silent mutation to disrupt the PAM motif. Strategy #2 aimed to KI mutation 19 bp upstream of the expected cutting site; another silent mutation introduced to disrupt the PAM motif using crRNA: cgtcatgtgcaggccaacag *agg and* repair template tttaggtccaaacatttggctagagtatggccagtactcagttggtggcattggtca gaaaggtggccttgagaaggttcgctctgtctttAaaagagccctgtcTtctgttggcctgca-catgacgaaaggcctggccatctgggaggcctaccgagagtttgaaagcgccatcgtggagg ctgctcgggtgagtccag.

To prepare the microinjection cocktail, crRNA was annealed with tracrRNA (IDT) to form duplex RNA, which was subsequently incubated with Cas9 protein (IDT) to form Cas9 Ribonucleoprotein. Repair template was then added into the cocktail. The final concentration of crRNA and tracrRNA was 15 ng/μL, with repair template at 10 ng/μL and Cas9 protein at 60 ng/μL. Microinjected C57BL/6 one-cell embryos were incubated overnight to the two-cell stage and surgically transferred to pseudopregnant CD1 females. For each targeting strategy, >250 embryos were microinjected, with >160 two-cell embryos transferred. Genotyping: the KI mutation creates a DraI site (tttAaa) not present in the wt. Primers 5'gggtgacatatcagtgagccgaactt3' and 5' agggccagtctcccagtcccactatc3' were used to amplify the target region.

## Statistics and reproducibility

Experiments were repeated independently yielding similar results the following number of times: Immunofluorescent staining of endogenous or transiently transfected WT or variant SART3 in HEK293t cells - three independent experiments with both SART3 and FLAG antibodies. *Drosophila* crosses and analyses - twice with each RNAi line. For *tj*-Gal4 crosses, these were carried out once with the dj-GFP reporter and twice without reporter for each RNAi line. Staining of human foetal testes and ovaries - once (on two different tissue samples per sex). Repeats were not possible due to scarcity of tissue. Staining and RNA analysis of gonadal organoid differentiation - three independent differentiation experiments were carried out with three biological replicate mono-layers/organoids analysed for each cell line at each timepoint. Neuronal staining experiments - five independent differentiation experiments were analysed, with only three providing sufficient numbers of homozygous variant neurons to analyse. Staining for MAP2, BIII, SART3 and

CC3 - three independent differentiations analysed, staining for ANK-G and Neurofilament - two. Western blotting for SART3 in iPSCs - three independent experiments. Immunofluorescent staining of iPSCs for SART3 and OCT4 – two independent experiments. Embryoid body experiments – two independent experiments. NT2/D1 KD experiments were replicated independently four times. For all other experiments where statistics were derived, details of the repeats, sample number and statistical methods can be found in the Figure Legends.

## Reporting summary

Further information on research design is available in the Nature Portfolio Reporting Summary linked to this article.

## Data availability

All data supporting the findings described in this manuscript are available in the article and its Supplementary Information files, and from the corresponding author upon request. The MS proteomic data generated in this study have been deposited in the ProteomeXchange Consortium via the PRIDE[90] partner repository under accession code PXD032816. RNA-seq data generated in this study have been deposited in the Sequence Read Archive (SRA) database under bioproject accession code PRJNA886829. *SART3* variants have been entered into ClinVar (SCV003842293 - SCV003842300). Exome sequencing data can be requested by contacting the corresponding author, with a response within 1 month. Due to ethics restrictions on storing and sharing our paediatric patient exome data, this data will have controlled access and will be limited to individuals who enter a research agreement. Use of this genomic data will be restricted to those named on the agreement, and exome data will be patient de-identified. The following databases were used; human reference genomes hg19 and hg38, Gencode human annotation release 40 (GRCh37), dbSNP 132, dbSNP 138, 1000 Genomes Project, NHLBI Go Exome Sequencing Project, gnomAD, TOPMed, ALFA, Flybase. All unique materials are readily available from authors upon request. Source data are provided with this paper.

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

## Acknowledgements

The authors would like to express sincere gratitude to the patients and their families for their participation in this study. Many thanks to the following facilities; The iPSC Derivation & Gene Editing Facility at the Murdoch Children's Research Institute who created the iPSC lines. They are supported by the Stafford Fox Foundation, Phenomics Australia (via the Australian Government through the National Collaborative Research Infrastructure Strategy programme) and the Novo Nordisk Foundation reNEW Center for Stem Cell Medicine (NNF21CC0073729). The Victorian Clinical Genetics Services (VCGS), Melbourne, Australia who carried out RNA sequencing. BPA-enabled (Bioplatforms Australia)/NCRIS-enabled (National Collaborative Research Infrastructure Strategy) infrastructure located at the Monash Proteomics and Metabolomics Facility, supported by Victorian State Government Operational Infrastructure Support funding. Bloomington *Drosophila* Stock Center (NIH P40OD018537) and the Vienna *Drosophila* Resource Center for fly stocks. The Australian *Drosophila* Biomedical Research Facility (OzDros). Many thanks to the

following people; Irene Ghobrial (MCRI) for her help and advice on growing iPSCs, and Dr Kiymet Bozaoglu (MCRI) for help with EB bodies. Prof. Johnny J. He (University of North Texas Health Science Center) for providing us with the shRNA-SART3 vector. Dr David Rollo for editing the manuscript. This study was supported by a National Health and Medical Research Council (NHMRC) programme grant (1074258) awarded to AS, NHMRC project grant (1156942) (K.A.), a Medical Research Future Fund Stem Cells Mission grant (MRF1201781) (K.A., B.N.R. and P.Kw), an Australian Research Council Future Fellowship (FT100100764) to M.B., A NHMRC Investigator Grant (1174040) to D.W., Agence Nationale de la Recherche funding ANR-10-LABX-73 REVIVE, ANR-17-CE14-0038-01 and ANR 20 CE14 0007 to K.M., ANR-19-CE140022 and ANR-19-CE14-0012 to A.B.; G.Z. and E.B. are members of the European Reference Network for Rare Neurological Diseases - Project ID No 739510.

## Author contributions

Conceptualisation: K.L.A., S.E., J.B., B.B.Z. and A.H.S. Methodology design: K.L.A., S.E., B.N.R., N.M.D., N.S., L.Z., J.B., K.R.S., R.B.S., C.Hu, G.R., J.v.d.B., F.C., S.P., G.R.H. and A.H.S. Experimental investigation, validation, and analysis: K.L.A., S.E., B.N.R., N.S., L.Z., J.B., R.B.S., C.Hu, G.R., J.v.d.B., F.C., J.C., S.P. and D.K.W. Bioinformatic analysis: S.E. and N.M.D. Genomic analysis and variant identification: S.E., K.S., K.W., G.Z., L.B., S.B.S., J.R., A.B. and K.M. Patient clinical notes and data curation: K.L.A., S.E., G.Z., A.R.R., N.G., C.H., Y.S., A.Z., E.Be, M.M., B.B.R., N.B., A.B., K.M., E.B., N.W. and B.B.Z. Resources: A.J., N.G. and C.H. Manuscript writing: K.L.A. and D.K.W. Manuscript editing: K.A., N.D., N.S., G.R., J.v.d.B., F.C., G.R.H. and A.H.S. Supervision: K.L.A., N.S., J.B., M.B., A.O., T.J.O.B., P.Kw, P.K., G.R.H. and A.H.S. Project administration: K.L.A., S.E. and A.H.S. Funding acquisition: K.L.A., B.N.R., T.J.O.B., P.Kw. and A.H.S.

## Competing interests

K.R.S. is currently an employee of AstraZeneca and own shares in the company. Their contribution to this article predated her employment at AstraZeneca. The remaining authors declare no competing interests.

## Additional information

[1]The Murdoch Children's Research Institute, Melbourne, Australia. [2]Department of Paediatrics, The University of Melbourne, Melbourne, Australia. [3]The Victorian Clinical Genetics Services, Melbourne, Australia. [4]Department of Neuroscience, Central Clinical School, Monash University, Alfred Centre, Melbourne, Australia. [5]Walter and Eliza Hall Institute of Medical Research, Melbourne, Australia. [6]School of BioSciences, Faculty of Science, University of Melbourne, Melbourne, Australia. [7]Department of Medical Biology, Faculty of Medicine, Dentistry and Health Sciences, University of Melbourne, Melbourne, Australia. [8]Department of Anatomy and Physiology, The University of Melbourne, Melbourne, Australia. [9]Institute for Molecular Bioscience, The University of Queensland, Brisbane, QLD, Australia. [10]School of Biomedical Sciences, The University of Queensland, Brisbane, QLD, Australia. [11]Genetics Institute, Rambam Health Care Campus, Rappaport Faculty of Medicine, Institute of Technology, Haifa, Israel. [12]Unit of Muscular and Neurodegenerative Disorders and Unit of Developmental Neurology, Department of Neurosciences, Bambino Gesù Children's Hospital, IRCCS, Rome, Italy. [13]Centre de Référence des Malformations et Maladies Congénitales du Cervelet, Et Laboratoire de Neurogénétique Moléculaire, Département de Génétique et Embryologie Médicale, APHP. Sorbonne Université, Hôpital Trousseau, Paris, France. [14]Developmental Brain Disorders Laboratory, Imagine Institute, INSERM UMR 1163 Paris, France. [15]Genetic Institute, Tel Aviv Sourasky Medical Center, Tel Aviv, Israel. [16]Genetics Institute, Kaplan Medical Center, Hebrew University Hadassah Medical School, Rehovot 76100, Israel. [17]Edmond and Lily Safra Children's Hospital, Chaim Sheba Medical Center, Ramat Gan, Israel. [18]Sackler School of Medicine, Tel Aviv University, Tel Aviv, Israel. [19]Department of Growth and Reproduction, Copenhagen University Hospital, Rigshospitalet, Copenhagen, Denmark. [20]Monash Proteomics and Metabolomics Facility, Biomedicine Discovery Institute, Department of Biochemistry and Molecular Biology, Monash University, Clayton, Australia. [21]The Peter MacCallum Cancer Centre, Melbourne, Australia. [22]School of Mathematics and Statistics, The University of Melbourne, Melbourne, Australia. [23]Department of Medicine, The Royal Melbourne Hospital, The University of Melbourne, Melbourne, Australia. [24]Aix-Marseille Université, APHM. Department of Pediatric Neurology, Timone Hospital, Marseille, France. [25]Radiology Department, Sheba medical Centre, Tel Aviv, Israel. [26]Kaplan Medical Center, Hebrew University Hadassah Medical School, Rehovot 76100, Israel. [27]Pediatrics Department, Kaplan Medical Center, Rehovot 76100, Israel. [28]Faculty of Medicine, Hebrew University of Jerusalem, Hadassah Medical School, Jerusalem, Israel. [29]Higher Institute of Nursing Sciences of Gabes, University of Gabes, Gabes, Tunisia. [30]Laboratory of Human Molecular Genetics, Faculty of Medicine of Sfax, Sfax University, Sfax, Tunisia. [31]Department of Congenital and Hereditary Diseases, Charles Nicolle Hospital, Tunis, Tunisia. [32]Institut Pasteur, Université de Paris, CNRS UMR3738, Human Developmental Genetics, 75015 Paris, France. [33]The Rina Mor Genetic Institute, Wolfson Medical Center, Holon 58100, Israel. [34]Pediatric Endocrinology Unit, Dana-Dwek Children's Hospital, Tel Aviv Medical Center, Tel Aviv, Israel. [35]Sheba Medical Center, Tel Aviv, Israel. [36]Present address: Department of Medical Biology, Faculty of Medicine, Dentistry and Health Sciences, University of Melbourne, Melbourne, Australia. ✉e-mail: katie.ayers@mcri.edu.au

