## [Peer Review File · Nature Communications]

Variants in SART3 cause a spliceosomopathy characterised by failure of testis development and neuronal defectsREVIEWER COMMENTS

Reviewer #1 (Remarks to the Author):

The authors have presented a fairly compelling that pathogenic variants in the Squamous cell carcinoma antigen recognized by T cells 3 (SART3) gene are associated with 46,XY gonadal dysgenesis (and other gonadal developmental disorders), intellectual disability, developmental delay, and brain abnormalities. The experimental work is quite strong, but the assembly of the manuscript is very uneven. It seems as though different sections were written by different authors with some having significant expository issues. Due care was not applied when assembling these sections. The medical genetics has many errors. The drosophila knockdowns, while important to the overall case, are hard to follow. The arguments for development mechanisms related to reduced expression of the FGF9 and NOVA2 genes are speculative, at best, and should be toned down.

Nonetheless, the finding of a splicesomopathy as a cause for these disorders is very novel and the identification of this phenotypic complex is important to the framing of differences of sex development as a potential complex genetic syndrome and not an isolated condition. Significant modification of this manuscript is required for it to be suitable for publication in this (or any) journal. A detailed critique follows.

Affiliations: inconsistent use of capitalization for institutional names

L74: Not all the subjects have 46,XY GD

L79: Not clear. The splicesosome and RNA splicing are not signaling pathways.

L84: Why are these a unique set of neuronal defects?

L93-94 Poor transition

L106: What is meant by an individualized management plan?

L114: Not clear why recent publications in the phosphoregulatory pathway are important to the arguments being made here.

L117: Do patients with Pontocerebellar hypoplasia type 7 have gonadal dysgenesis? Is the example germane?

L130: What sort of developmental delays? The question of whether the affected subjects have a consistent set of neurologic findings as features of their genetic syndrome is important.

L141: Many were 46,XX and did not have gonadal dysgenesis.

L145: What was the relatedness?

L155: The term “carrier” has fallen out of favor in genetic circles. Were the sibs in all of the families heterozygotes?

L169: Was the hormonal profile based on an HCG stimulation test?

L170: For ISR2.2, the FSH was reported, “normal” in Table 1 at 2 months. This does not seem to be hypogonadotropic hypogonadism. In any case, the diagnosis could be made only after an LRH test.

L177: ISR2.2 is at what age for pubarchy to be informative? Also use of past tense is inconsistent.

L200: In silico predictions are meaningless.

L206: Not a sentence.

L249-42: What are traffic jam and don-juan?

L288: Does staining for SART3 indicate a requirement? I don't think so.

L296: What does fewer internal structures mean?

L302: Contradicts L296 “developed structures” versus “fewer internal structures”?

L386: Is reduction of FGF9 compelling evidence? Wouldn't there be disruption of the SOX9 feedback loop?

L433: What is the evidence for high pre-mRNA turnover?

L442: How do overlapping neurologic findings different from progressive neurodegenerative disease?

L489: Cancer association seems exaggerated.

Figure 3 legend is unclear. Are these drosophila embryos? There should be better labels.

Minor suggestions:

Check that authors are following HGV nomenclature.

Capitalization of names is inconsistent.

Moroccan Jews are not a control for North African (Tunisian?) Jews.

Reviewer #2 (Remarks to the Author):

The manuscript by Ayers and co-authors reports the identification of recessive mutations in the SART3 gene that are associated with gonadal dysgenesis, brain development abnormalities and intellectual disability. The authors recapitulate similar developmental defects in vivo by knocking down the homologue of SART3 in *Drosophila* (Rnp4f) and ex-vivo by introducing some of the identified mutations in human iPSCs and inducing their differentiation in testicular organoids or neuronal lineage. Lastly, the authors perform high-throughput transcriptomic and proteomic analyses to identify the genes that are altered in undifferentiated iPSCs harboring the SART3 mutation. Collectively, they study is of broad interest because it identifies novel mutations that are directly linked to a genetic syndrome in multiple families. This finding paves the ground for evaluation of SART3 mutations in children displaying 46XY karyotype and gonadal dysgenesis at birth, possibly improving clinical management of these patients. While the clinical and pathological characterization of the patients is well described, the molecular and mechanistic support to the hypothesis appears more preliminary.

Major comments

- 1) In Figure 3 and Supplementary Figure 3, no data were shown to evaluate the efficiency of RNAi transgenes against Rnp4f in *Drosophila* and to show the extent of knockdown of Rnp4f.
- 2) Analysis of RNA-seq data identified 4765 differentially expressed genes in homozygous SART3.p.Arg836Gln iPSC lines respect to control lines (2487 genes were downregulated and 2278 were upregulated). Gene ontology analysis of the top 1000 most deregulated genes (Figure 6b) highlighted categories like signaling and the spliceosome, which may be altered in homozygous SART3.p.Arg836Gln iPSC lines. The authors should also test gene ontology analyses for all the deregulated genes, as using 1000/4765 genes may not be representative. It would also be useful to carry out the analysis separately for up- and for down-regulated genes, thus highlighting the type of regulation for the enriched functional classes.
- 3) The authors should perform validation of the transcriptomic and proteomic analyses by RT-PCR and Western blot analyses on a separate sample set. A reasonable number of target genes, splice variants and proteins should be analyzed and validated.
- 4) The authors propose that there is a disruption of the spliceosome in presence of SART3 variant, even though most of the transcripts/proteins deregulated and represented across the spliceosome were up-regulated (Figure 6c). The interpretation of this result is not clear. The global splicing analysis did not show a loss of splicing efficiency in SART3 variant-carrying cells and alternative splicing events were described only superficially. General disruption of splicing efficiency can be measured onnascent

labelled transcripts in mutated iPSC cells. This type of analysis is necessary to support the claim. Moreover, it seems that SART3 mutation more globally affects gene expression rather than splicing. A more in-depth analysis of the molecular phenotype of the mutated cells and/or of NT2 cells depleted of SART3 should be carried out.

5) Interestingly, homozygous SART3.p.Arg836Gln iPSC lines and NT2/D1 cells knocked down for SART3 expressed low levels of the splicing factor NOVA2, which may explain some of the neuronal phenotypes. Unfortunately, neither the mechanism that correlates the expression of SART3 and NOVA2, nor if NOVA2 downregulation is functional to the changes in alternative splicing observed in SART3 mutant cells (i.e. SART3 Arg836Gln variant), was investigated. The authors should perform these analyses to support their claim of genetic interaction between the two genes.

Minor comments

1) The authors did not report whether children born with the recessive SART3 variants also harbored other known causative genetic variants.

2) The conclusion that SART3 variants do not affect subcellular localization is poorly supported. Fluorescence analysis alone is not sufficient to discriminate a different localization within the nucleus.

3) The conclusion that patient SART3 variants are incompatible with viability in mice is poorly supported. Only two (Arg836Gln and Glu211Lys) variants were investigated. It is necessary to clarify why only these two variants have been tested.

4) In the Materials and Methods section, the abbreviations are not clear (for example PBTX).

5) In the section "Induced pluripotent stem cells carrying SART3 patient variants show disrupted differentiation into testis and neural cell lineages", quantification of immunostaining for the various neuronal markers would help the interpretation of the data.

6) The Edu incorporation experiment for proliferation assay is not described in Materials and Methods section.

7) Check the numbering of the supplementary tables because there is no correspondence in the text.

8) Correction in the text at the line 357: “Fisher’s exact test”

9) In the Figure 2d,e and in Supplemental Figure 2a,b, the western blot analysis of WT and variant pCMV-SART3-FLAG constructs transiently transfected in HEK293t cells were shown. The authors suggested that transfection of Ser216Pro, Arg493Trp, Arg519Gly and Pro718L variants shown a reduced protein expression of SART3 and Arg253 variant was subject to nonsense mediated decay. However, statistical significance was not shown. Moreover, if the mutations were introduced into a cDNA, how can the transcript without introns and exon-exon junctions be degraded by NMD?

Additional experiments are needed to prove that nonsense mediated decay occurs. The NMD substrates are stabilized by the translation inhibitor cycloheximide (CHX). To verify the involvement of the NMD pathway in the regulation of SART3 variant level, transfected cells can be treated with CHX to analyze the overexpression of transcripts underwent NMD.

The representative western blot image and their relative band intensity histograms would be more useful if they followed the same order.

10) Line 345: the authors state “Cajal bodies, the site of spliceosome activity”. Cajal bodies are NOT the site of spliceosome activity. The spliceosome functions on the chromatin and most introns are spliced co-transcriptionally. Splicing factors accumulate in splicing speckles (not Cajal bodies) and are recruited to nascent transcripts on the chromatin.

11) Line 430 the authors state “This effect may be exacerbated in tissues such as the testis and CNS with high proliferation rates or which rely heavily on splicing”. Testis and CNS are not organs with high proliferation rates. The CNS is made of mostly post-mitotic neurons; in the testis, the only actively proliferating cells are the spermatogonia. Thus, the statement is incorrect. Regarding the dependency on splicing of these two organs, the authors should refer to specific studies that indicated it.

Reviewer #3 (Remarks to the Author):

In this manuscript, Ayers et al reported bi-allelic variants in SART3 gene from six families with children demonstrating 46,XY gonadal dysgenesis, developmental delay and intellectual disability. Further, all patient variants affect highly conserved amino acids in protein domains essential for SART3 function. Using six families and CRISPR/Cas9-engineered iPSC, they further examined and found that one disease-causing mutant (p.Arg836Gln) exhibit disrupted differentiation into testis and neural cell lineages. By using RNA sequencing and LC-MS/MS, in the differentially expressed gene, cancer signaling and stem cell pluripotency gene are significantly enriched as well as the spliceosome genes. And a highly downregulated genes NOVA2 which may lead to neurodevelopmental disease through interacting with

SART3. Nevertheless, SART3 protein is not mis-localized in p.Arg836Gln iPSCs. And the introduction of SART3 patient variants in mouse models results in embryonic lethality. Consistent with the results in the mutated iPSCs, knock down of Rnp4f, the SART3 orthologue of fly, also resulted in the infertile of male flies and fail of normal neural development during embryogenesis.

Overall, this is a well-organized manuscript with significant study that identifies many pathogenic variants in SART3, and their effects to gonadal and neural development. This work is also important for understanding spliceosomopathy caused by SART3 in 46,XY DSD patient. However, several concerns should be addressed.

Major concerns:

1. Among those bi-allelic variants in SART3 gene, why did the authors choose p.Arg836Gln as the study target, why not other variants? For example, the variant in the TUN1.1 family.
2. Instead of using the KD Drosophila strain for investigation, why didn't make a point mutant strain according to the SART3 mutations in human patients?
3. Since SART3 has two RRM for interaction with U6 snRNA, and the Arg836Gln mutation caused significantly changed expression of spliceosome genes in the iPSCs, it is necessary to analyze their RNA-seq data to see how much splicing events were differentially changed.

Minor:

1. Size of the Table 1 should be adjusted.
2. For many figures, labelling of panel letters and scale bars are not in consistent ways.
3. For scales, the "um" should be "µm".

Reviewer #4 (Remarks to the Author):

In their manuscript "Variants in SART3 cause a novel spliceosomopathy characterized by failure of testis development and neuronal defects" by Ayers and colleagues focusses on the role on SART3 variants in failures related to testicular development and neuronal defects. The authors identified recessive variants in the SART3 in 9 patients with 46,XY gonadal dysgenesis, intellectual disability, developmental delay and brain anomalies. The conserved role of SART3 was studied in knockdown of the Drosophila orthologue of SART3, due to the fact that null and knock-in patient alleles were lethal in mice. Furthermore, in vitro differentiation of hiPSCs into gonadal and neuronal cells was used to study the role of SART3 variants on testicular and neuronal development.

General comments:

Overall, the study is well-written and deals with an interesting aspect related to failure of the male reproductive organ and neuronal defects. The authors employed several experimental strategies to elucidate the role of SART3 variants, which are difficult to study, especially in humans. However, to improve the content of the current work and to increase the benefit for the reader, I suggest to consider my comments listed below.

Specific comments:

Results:

In figure 1A; please add the karyotype of the male offspring of the "ISR3-couple", or add the information, why the karyotyping has not been performed.

The histological pictures shown in figure 1b-g are not suitable for publication, due to low quality. Higher magnification as well as improved picture quality is needed to appreciate the results described in the text. In addition, to higher magnifications, overview pictures with the same magnification need to be added.

Page 8, Line 162: Please specify "evidence of some early testicular function" more in detail. What does "some" include?

Page 8, Line 168: The mentioned “small areas of seminiferous tubules” are not clearly shown in figure 1b-g (see also comment above). Please revise figure 1 and improve the picture quality.

Page 8, Line 171: Please add reference values for the gonadotropins measured, to allow the reader to appreciate, what is meant by “low (levels of) gonadotropins”. In this respect, I strongly suggest to add data from healthy controls or reference values (e.g., hormone reference data, frequency of DSDs mentioned) to allow the reader to appreciate the full data set.

Page 11, Line 249: Please describe the effect on KD on the somatic cell compartment mentioned, more in detail. Which somatic cells are affected? Is there a direct effect on the germ cells or via the somatic environment building the stem cell niche? Is there any connection to the Sox9 (a Sertoli cell marker also mentioned in the introduction of the manuscript in another context)?

Page 12, Line 276 & 279: Please add the information regarding the species, here.

Page 13, Line 282 – 283: Please add the details regarding the markers and expression profile (e.g., RNA, DNA or protein level) as supplementary data.

Figure 4f-n: Please add higher magnifications here, as well.

Page 13, Line 290-291: Please add the information regarding the functionality tests used for these organoids. As it appears from the text and figures, the similarities to the organoids described in Knarston et al. 2020 are rather limited.

Page 13, Line 303-304: The results described are not enough to confirm the statement made, since all markers are also expressed in KD, and structural features can vary under in vitro conditions. In order to confirm the results, the functionality of the organoids need to be demonstrated (see also comment above).

Suppl. Figure 4d shows morphological impairments/differences. Therefore, tests of used iPSCs, incl. teratoma-formation assay, in addition, to details of marker panels used for the different cells need to be added.

In this respect, the effect of SART3 variants on the undifferentiated iPSCs needs to be evaluated. How much do cell line specific features affect the ability of the cells to differentiated into gonadal and neuronal cells?

Page 14, Line 323-324 and related to suppl figure 4 & 5; fig.4: What is the effect of SART3.p.R836Q Het in terms of gonadal cell differentiation?

Page 17: The authors mention that NT2/D1 is cell line used as model for somatic Sertoli cells. However, this cell line is a multipotent cell line derived from human embryonal carcinoma cells (a rather uncommon germ cell tumor type) and therefore not really optimal to model somatic cells, such as Sertoli cells. Therefore, additional references should be included, showing the suitability of the cell line for the performed studies.

Page 17, Line 382-383: No effects on genes important for testicular development, such as NR5A1, WT1 and SOX9 were observed, which is very interesting as those include the main players, such as SOX9 for early male gonadal formation. Please discuss this as well as the effect on FGF9 in relation to the before mentioned genes more in detail in the discussion.

Material and methods:

Please add the information regarding all negative and positive controls used (incl. samples, antibodies, concentrations) for the different staining performed in the study used to validate the antibodies used.

We thank the four reviewers for their comments and questions and have provided point-by-point responses below. Please note: line numbers correspond to those in the clean version without tracked changes.

REVIEWER COMMENTS

Reviewer #1 (Remarks to the Author):

The authors have presented a fairly compelling that pathogenic variants in the Squamous cell carcinoma antigen recognized by T cells 3 (SART3) gene are associated with 46,XY gonadal dysgenesis (and other gonadal developmental disorders), intellectual disability, developmental delay, and brain abnormalities.

The experimental work is quite strong, but the assembly of the manuscript is very uneven. It seems as though different sections were written by different authors with some having significant expository issues. Due care was not applied when assembling these sections. The medical genetics has many errors. The drosophila knockdowns, while important to the overall case, are hard to follow. The arguments for development mechanisms related to reduced expression of the FGF9 and NOVA2 genes are speculative, at best, and should be toned down.

Nonetheless, the finding of a splicesomopathy as a cause for these disorders is very novel and the identification of this phenotypic complex is important to the framing of differences of sex development as a potential complex genetic syndrome and not an isolated condition. Significant modification of this manuscript is required for it to be suitable for publication in this (or any) journal. A detailed critique follows.

We thank reviewer 1 for their critique and have now substantially edited the paper to achieve consistency between sections. We have addressed medical genetics issues. In addition, we have rewritten the Drosophila section and hope it is now easier to follow. We agree with the comment considering FGF9 and NOVA2 to be a disease mechanism – this has been toned down throughout the text.

1. Affiliations: inconsistent use of capitalization for institutional names

Thank you, we have corrected this.

2. L74: Not all the subjects have 46,XY GD.

Agreed: we have changed this to read “ we have identified recessive variants in the SART3 gene in nine individuals presenting intellectual disability, developmental delay and brain anomalies and gonadal dysgenesis in those with a 46,XY chromosome complement. We have defined this better throughout the manuscript.

3. L79: Not clear. The spliceosome and RNA splicing are not signaling pathways.

Agreed: we have removed the reference to these now and have included mention that spliceosome components are upregulated. Line 79 “demonstrated disruption to multiple signalling pathways and upregulation of spliceosome components.”

4. L84: Why are these a unique set of neuronal defects?

We have removed “unique”. Instead we now focus on the overlapping clinical presentation in the patients.

5. L93-94 Poor transition

Thank you we have changed this.

6. L106: What is meant by an individualized management plan?

We have reworded this to read “An early molecular diagnosis is clinically beneficial for individuals born with a DSD as it may inform patient clinical management in relation to adrenal and gonadal function, gender development, and gonadal cancer risk” Line 100.

7. L114: Not clear why recent publications in the phosphoregulatory pathway are important to the arguments being made here.

This has been removed.

8. L117: Do patients with Pontocerebellar hypoplasia type 7 have gonadal dysgenesis? Is the example germane?

Several publications have found gonadal abnormalities such as vanishing testis/gonadal dysgenesis in PCH-7. In the Lardelli study (2017) phenotypes “ranging from absent gonads to ovarian and uterine remnants or atrophic and undescended testes” were described. Therefore we believe the example is germane, and we have clarified this in the text stating “a syndrome characterized by neurodegeneration and abnormalities of the gonads or external genitalia”.

9. L130: What sort of developmental delays? The question of whether the affected subjects have a consistent set of neurologic findings as features of their genetic syndrome is important.

We have now added in more detail regarding both the severity of ID and the extent and severity of developmental delay where possible. In addition to this, detailed clinical observations are provided in Supp. Table 1.

The text now reads LINE 185 “All children had intellectual disability (ID) reported, ranging from mild to profound, and hypotonia. All were reported to have global developmental delay with most showing delays in all aspects including motor skills, speech and language, communication and socioemotional skills” (Table 1, see Supp. Table 1 for detailed clinical notes).

10. L141: Many were 46,XX and did not have gonadal dysgenesis.

Agreed, we have now clarified this throughout the manuscript.

11. L145: What was the relatedness?

Supplementary Table 2 shows the proportion of alleles shared identical by descent (PI_HAT) between parents and children from ISR1 and ISR2, both within and between families. We have changed the legend here to make the comparisons easier to understand in conjunction with the pedigrees in Figure 1. PI-HAT figures vary between comparisons, but inter-family comparisons of children between ISR1 and ISR2 fall between 0.236 – 0.246. Comparisons of parents between families is between 0.185-0.242. Given limited space in the main text we have not included these details, but hope they are now easily accessed in the Supp Table 2.

12. L155: The term “carrier” has fallen out of favor in genetic circles. Were the sibs in all of the families heterozygotes?

We have removed reference to carriers. All affected siblings (in ISR1, ISR2, TUN1) were confirmed to be homozygous for SART3 variants. Genetic testing was only carried out for unaffected siblings in families in ISR2 (they are heterozygous). Unfortunately, genetic testing is not available for the unaffected siblings in ISR3 and FRA1 as parents did not wish to test these children. This has now been added to the text and indicated in Figure 1.

13. L169: Was the hormonal profile based on an HCG stimulation test?

Thank you for this question. The endocrinological testing was carried out for this child at 2 days of age, and again at 2 months of age, during mini-puberty. Not, as we had suggested, at the time of surgery – this has been corrected. At 2 months of age, the child had undetectable basal testosterone, and a very low response to hCG stimulation (0.6 where >7 is considered a good response). We had included all details and reference ranges in the Supp Table due to text constraints but have now put these into Table 1 for clarity.

14. L170: For ISR2.2, the FSH was reported, “normal” in Table 1 at 2 months. This does not seem to be hypogonadotropic hypogonadism. In any case the diagnosis could be made only after an LRH test.

At 2 months of age the combination of low basal testosterone, poor response to hCG stimulation and low AMH support gonadal dysgenesis (primary hypogonadism), which is usually coupled with normal to high levels of FSH and LH. Given that these were low (LH) or normal (FSH), this indicates secondary hypogonadism as well. We have clarified this in the text now. LH and FSH levels and reference ranges have now also been added to Table 1.

15. L177: ISR2.2 is at what age for pubarchy to be informative? Also use of past tense is inconsistent.

ISR2.2 was born in 2008 (is currently aged 14). Although can we assume that the reviewer meant ISR1.2 the 46,XX child? This child was 16 at last clinical visit. We have included this in the text now. Tenses have been corrected.

16. L200: In silico predictions are meaningless.

While we generally agree, many researchers still consider these during their analyses and we have found in the past when we have omitted these they have been asked for during revision. Therefore we have left them in Table 2 but have reduced reference to them. LINES 207. "In-silico tools such as Polyphen-2, PROVEAN, SNAP2, muPRO, SIFT, predict variants to be mostly pathogenic/damaging"

17. L206: Not a sentence.

Corrected

18. L249-42: What are traffic jam and don-juan?

traffic-jam is the gene controlling a Gal4 driver for transgene expression and don-juan is a reporter line where GFP is under the control of this gene. We have rewritten this passage to elaborate and better describe the Drosophila work.

19. L288: Does staining for SART3 indicate a requirement? I don't think so.

Agreed, we have removed this.

20. L296: What does fewer internal structures mean?

L302: Contradicts L296 "developed structures" versus "fewer internal structures"?

Thank you for this observation. We have added more data to this section, including additional images and gene expression. We find that the homozygous variant organoids are smaller with more apoptosis, and that the tubular structures are generally unaffected. What we meant is that the apoptosis appears to affect the internal regions outside of the tubular structures – but have clarified our findings in the text now. We also find differences in the expression of gene markers of gonadal differentiation – also added to Figure and text.

21 L386: Is reduction of FGF9 compelling evidence? Wouldn't there be disruption of the SOX9 feedback loop?

Yes this is an interesting and highly valid point. We have found that FGF9 is reduced in the NT2D1 SART3 KD, and also in the organoid differentiation, where it is usually activated at day 7, but where the SART3 variant cells show little to no activation. In both cases, SOX9 expression is not affected. We do not have an explanation for this, however, believe it is important to include as it shows that SART3 loss or variation does affect key molecules in the testis determination pathway. And, as we do not have an effective FGF9 antibody, we have been unable to demonstrate that this loss of FGF9 mRNA translates to a reduction at the protein level, and so this may explain how SOX9 and other Sertoli markers are still present.

22 L433: What is the evidence for high pre-mRNA turnover?

Thank you for this comment. In this reference the authors looked at in vitro splicing efficiency in the absence of SART3 and found that SART3 was only required for splicing in conditions where they added in “high levels” (5 or 10 fold higher than normal conditions) of pre-mRNA, concluding that “these findings demonstrate that p110 is required as a splicing factor only under conditions when relatively high levels of pre-mRNA are turned over, consistent with recent evidence that p110 is involved in U4/U6 snRNP recycling.” Our reason for including reference to this work as a discussion how variants in SART3 might affects some tissues more than others. In particular, the CNS and testis which have appear to have higher levels of splicing (which we have now referenced). We agree that this is different to “high pre-mRNA turnover” and have now adjusted the text to be more specific:

LINES 500-504 In vitro studies have suggested SART3 is needed as a splicing factor specifically in conditions where there is an increased requirement for the spliceosome¹⁸. Indeed, the brain and testis largely exceed the other tissues with respect of splice-variants expression⁵⁶⁻⁵⁹, and higher expression of specific signatures of splice factors suggests that alternative splicing plays a particularly crucial role in these tissues⁶⁰.

23. L442: How do overlapping neurologic findings different from progressive neurodegenerative disease?

They differ in that the neurologic conditions described in our patients have presented early in childhood, and there is no evidence for neurodegeneration either clinically or in imaging (i.e. the neurological disorder does not seem progressive). We have now reworded this to read LINES 436-440 “All children described here had severe neurodevelopment symptoms from infancy, including intellectual disability, global developmental delay and neurological anomalies including agenesis/hypoplasia of the corpus callosum, enlarged ventricles and cerebellar atrophy. The overlapping findings are not stereotypical of other known conditions and there is no evidence of progressive neurodegeneration clinically or in imaging in any of the patients.”

24. L489: Cancer association seems exaggerated.

We agree. As we are unable to get extended family histories for all the pedigrees, we have now removed mention of these cases, and have just made reference to the fact that SART3 variants have been implicated in malignancies in the literature, and that families may benefit from additional monitoring.

25. Figure 3 legend is unclear. Are these drosophila embryos? There should be better labels.

Figure 3 legend and the results have now been rewritten to be clearer.

Minor suggestions:

26. Check that authors are following HGVS nomenclature.

Thank you we have now corrected this.

27. Capitalization of names is inconsistent.

Thank you we have now corrected this.

28. Moroccan Jews are not a control for North African (Tunisian?) Jews.

ISR1 and ISR2 are actually of Moroccan-Libyan Jewish descent living in Israel, hence the inclusion of a screen of 165 individuals of Moroccan Jewish ancestry as a control. We have changed this in the text now to be more specific about the North African region.

Given that each family in this study is from a different ethnic background and that Jewish history is complex, it has been difficult to collectively screen for SART3 variants in a large group of appropriate controls for each family. Nevertheless, from our investigations into the frequency of all variants in numerous extensive online databases including GnomAD (where all variants were rare or absent), and through several screening efforts provided by the individual clinics/labs who first identified each variant (including the North African population, and a large cohort of Jewish individuals – information now added to the text) we are confident that these SART3 variants are very rare in the general unaffected population.

Reviewer #2 (Remarks to the Author):

The manuscript by Ayers and co-authors reports the identification of recessive mutations in the SART3 gene that are associated with gonadal dysgenesis, brain development abnormalities and intellectual disability. The authors recapitulate similar developmental defects in vivo by knocking down the homologue of SART3 in Drosophila (Rnp4f) and ex-vivo by introducing some of the identified mutations in human iPSCs and inducing their differentiation in testicular organoids or neuronal lineage. Lastly, the authors perform high-throughput transcriptomic and proteomic analyses to identify the genes that are altered in undifferentiated iPSCs harboring the SART3 mutation. Collectively, their study is of broad interest because it identifies novel mutations that are directly linked to a genetic syndrome in multiple families.

This finding paves the ground for evaluation of SART3 mutations in children displaying 46XY karyotype and gonadal dysgenesis at birth, possibly improving clinical management of these patients.

While the clinical and pathological characterization of the patients is well described, the molecular and mechanistic support to the hypothesis appears more preliminary.

Major comments

29. 1) In Figure 3 and Supplementary Figure 3, no data were shown to evaluate the efficiency of RNAi transgenes against Rnp4f in *Drosophila* and to show the extent of knockdown of Rnp4f.

We agree this is important. We used two different transgenic RNAi lines against Rnp4f. In Supp Figure 3f (table) we had provided the relative knockdown efficiency of these lines (which was 70% KD for RNAi-B and 51% for RNAi-V). This was determined using digital droplet PCR on adult flies expressing the RNAi under the tubulin-Gal4 driver compared to controls, as detailed per Supp Materials and Methods. We have now changed the table so this is clearer.

30. 2) Analysis of RNA-seq data identified 4765 differentially expressed genes in homozygous SART3.p.Arg836Gln iPSC lines respect to control lines (2487 genes were downregulated and 2278 were upregulated). Gene ontology analysis of the top 1000 most deregulated genes (Figure 6b) highlighted categories like signaling and the spliceosome, which may be altered in homozygous SART3.p.Arg836Gln iPSC lines. The authors should also test gene ontology analyses for all the deregulated genes, as using 1000/4765 genes may not be representative. It would also be useful to carry out the analysis separately for up- and for down-regulated genes, thus highlighting the type of regulation for the enriched functional classes.

We agree this is a good idea. Initially we used only the top 1000 DE genes, and not the full DE list, as DAVID is optimised to run on lists less than 3000 genes and its Functional Annotation Clustering and Gene Functional Classification have a 3000 gene limit.

We did not split genes into up- or down- regulated as we were interested in capturing disrupted signalling pathways including both positive and negative regulators which may be affected differently. Nevertheless, we have now included annotation of the full 4765 gene list, as well as the up-regulated and downregulated genes in the Supp Table 5. We have gained some valuable insight from this including the observation that upregulated genes from the full set appear to represent KEGG pathways such as cancer signalling, pluripotency, organ development, spliceosome and ribosome as found in previous analysis. Those that are downregulated commonly implicated in metabolic pathways. This has now been added to the LINES 367-370 "Analysis of just up or downregulated DE genes found upregulated genes were enriched for terms such as regulation of transcription, development and axon guidance, whereas downregulated genes were enriched for metabolism terms among others (Supp. Table 5)."

We also performed this kind of analysis for the proteomic data (now found in Supp Table 5).

31. 3) The authors should perform validation of the transcriptomic and proteomic analyses by RT-PCR and Western blot analyses on a separate sample set. A reasonable number of target genes, splice variants and proteins should be analyzed and validated.

We have now included validation in the Supp data (Figure 6). This includes validation of a mix of up- and down regulated genes from the RNA-seq (including SART3, NOVA2, SIX3, UTF1, PRPF3, DDX23 and VAT1L by qPCR in three independent sets of iPSCs for each genotype. For validation of mass spectrometry data we have included western blot validation for three proteins (SART3, OCT4 and AASS). For alternative splice isoforms we have included validation of three genes which show DTU. Raw data or uncropped blots for these experiments has also been added to the source file.

32. 4) The authors propose that there is a disruption of the spliceosome in presence of SART3 variant, even though most of the transcripts/proteins deregulated and represented across the spliceosome were up-regulated (Figure 6c). The interpretation of this result is not clear.

The global splicing analysis did not show a loss of splicing efficiency in SART3 variant-carrying cells and alternative splicing events were described only superficially. General disruption of splicing efficiency can be measured on nascent labelled transcripts in mutated iPSC cells. This type of analysis is necessary to support the claim.

Moreover, it seems that SART3 mutation more globally affects gene expression rather than splicing.

A more in-depth analysis of the molecular phenotype of the mutated cells and/or of NT2 cells depleted of SART3 should be carried out.

We agree with the statement that SART3 mutation more globally affects gene expression rather than splicing. Indeed, we have now carried out rMAT analysis of the RNA-seq data which has indicated that there was no significant difference in the numbers of splicing events between the control, heterozygous and homozygous sart3 iPSC lines (i.e. the numbers of intron retention, mutually exclusive exons, exon skipping and alternative 5' or 3' splice site events were similar in the different cell lines). We have now included reference to these findings and have changed the title of this section to "SART3 variant iPSCs have widespread changes to gene expression and signalling". We have clarified in the text that SART3 variants do not appear to result in a significant increase in splicing events, rather, we see changes in general gene expression and in transcript usage for genes. We have also clarified that in general we see upregulation of spliceosome components. We postulate that this represents a compensatory mechanism in response to reduced SART3 levels or activity, and that this may also be the reason why major global disruption to splicing is not observed. It is also possible that splicing disruption would only be observed in

certain cell types or tissues, specifically patient cells or fetal tissues during development. We have also edited the discussion to reflect these changes.

LINES 380-386. rMATS analysis³⁰ did not find significant differences in the number of splicing events such as exon skipping or intron retention between control and SART3 homozygous variant iPSCs (data not shown), but differential transcript usage (DTU) analysis using equivalence classes³¹ found 347 transcripts from 232 genes with DTU between control and homozygous cells (FDR <0.05) (Supp Table 5). Thus changes in gene expression and differential transcript usage are associated with the homozygous SART3 variant in the pluripotent state. “

33. 5) Interestingly, homozygous SART3.p.Arg836Gln iPSC lines and NT2/D1 cells knocked down for SART3 expressed low levels of the splicing factor NOVA2, which may explain some of the neuronal phenotypes.

Unfortunately, neither the mechanism that correlates the expression of SART3 and NOVA2, nor if NOVA2 downregulation is functional to the changes in alternative splicing observed in SART3 mutant cells (i.e. SART3 Arg836Gln variant), was investigated. The authors should perform these analyses to support their claim of genetic interaction between the two genes.

We agree that whilst we have seen NOVA2 downregulation in both the iPSC model and in the NT2D1 model, that further investigation into the interaction between these genes is warranted. However given the current breadth of data in this paper we believe this is outside of the scope of this study, and we have reduced the reference to NOVA2 as a potential disease mechanism in the text accordingly.

Minor comments

34. 1) The authors did not report whether children born with the recessive SART3 variants also harbored other known causative genetic variants.

All patients underwent exome sequencing and no pathogenic/diagnostic variants in genes that could explain the condition were identified. ISR1 and ISR2 also had CMA, MECP2, CDKL5, FOXG1, DAX1, L1CAM and ARX ruled out as the causative genes using Sanger sequencing and Copy Number Variant (CNV) arrays (Illumina Omni 2.5) as detailed per the materials and methods. We have now clarified this lack of additional gene variants in the text for all patients.

35. 2) The conclusion that SART3 variants do not affect subcellular localization is poorly supported. Fluorescence analysis alone is not sufficient to discriminate a different localization within the nucleus.

Agreed, we have clarified this in the text, which now reads LINE 230 “Immunostaining showed that all missense variants demonstrated nuclear localisation similar to endogenous SART3 and transiently expressed WT SART3 (Figure 2f).

36. 3) The conclusion that patient SART3 variants are incompatible with viability in mice is poorly supported. Only two (Arg836Gln and Glu211Lys) variants were investigated. It is necessary to clarify why only these two variants have been tested.

Yes we agree that by only investigating the knock-in of two patient variants we cannot definitively say that all SART3 patient variants are incompatible with viability. We have removed this sentence from the abstract. We have also clarified in the text why we used these two variants and have changed our conclusion.

Lines 289 “ as these were the first variants identified, fell within different protein domains and had strong genetic evidence for causation.”

LINE 295 “We concluded it likely that these patient SART3 variants are incompatible with viability in mice

37. 4) In the Materials and Methods section, the abbreviations are not clear (for example PBTX).

We have now corrected this.

38. 5) In the section “Induced pluripotent stem cells carrying SART3 patient variants show disrupted differentiation into testis and neural cell lineages”, quantification of immunostaining for the various neuronal markers would help the interpretation of the data.

Thank you for this comment. Differentiation of the homozygous variant into cortical neurons was very inefficient, often failing or resulting in very few neurons compared to the heterozygous or homozygous controls. Given the vastly reduced number of neurons and reduced networks, we don't believe that quantification of markers would be informative due to differences of cell numbers and antibody availability/concentration per cell. Instead, we have provided additional images, including low magnification images to illustrate networks, and images for the heterozygous cells in Figure 5, and we hope this demonstrates our observations more clearly.

39. 6) The Edu incorporation experiment for proliferation assay is not described in Materials and Methods section.

Apologies, this has now been added to the supplementary file Materials and Methods.

40. 7) Check the numbering of the supplementary tables because there is no correspondence in the text.

This has been corrected.

41. 8) Correction in the text at the line 357: “Fisher’s exact test”

This has been corrected

42. 9) In the Figure 2d,e and in Supplemental Figure 2a,b, the western blot analysis of WT and variant pCMV-SART3-FLAG constructs transiently transfected in HEK293t cells were shown. The authors suggested that transfection of Ser216Pro, Arg493Trp, Arg519Gly and Pro718L variants shown a reduced protein expression of SART3 and Arg253 variant was subject to nonsense mediated decay. However, statistical significance was not shown. Moreover, if the mutations were introduced into a cDNA, how can the transcript without introns and exon-exon junctions be degraded by NMD? Additional experiments are needed to prove that nonsense mediated decay occurs. The NMD substrates are stabilized by the translation inhibitor cycloheximide (CHX). To verify the involvement of the NMD pathway in the regulation of SART3 variant level, transfected cells can be treated with CHX to analyze the overexpression of transcripts underwent NMD.

We have now carried out statistical analysis on relative band intensity from three independent experiments and have not found statistically significant reduction for any variant aside from the truncating variant 253. We have now reworded the text to reflect this.*

LINES. 225-230 “stability, FLAG-tagged SART3 missense variants were transiently transfected into HEK293T cells. Western blot analysis found a significant loss of signal for the p.Arg253 variant (Figure 2d,e, Supp. Figure 2a,b). Also Ser216Pro, p.Arg493Trp, Arg519Gly and Pro718Leu variants had a reduced signal with both FLAG and SART3 antibodies, although this did not reach statistical significance over three replicates (Figure 2d,e, Supp. Figure 2a,b). “*

In addition, it was an error on our part to discuss NMD, where we meant to say the truncating variant was not detected. This has been corrected.

43. The representative western blot image and their relative band intensity histograms would be more useful if they followed the same order.

Thank you for this useful comment. We have adjusted the order in both the main figure and supp figure to reflect the blots shown. Raw data and uncropped blots have now also been included in the Source Data file

44. 10) Line 345: the authors state “Cajal bodies, the site of spliceosome activity”. Cajal bodies are NOT the site of spliceosome activity. The spliceosome functions on the chromatin and most introns are spliced co-transcriptionally. Splicing factors accumulate in splicing speckles (not Cajal bodies) and are recruited to nascent transcripts on the chromatin.

Thank you for this expert guidance. We have revised this to now read “a nuclear structure important for snRNP maturation and where SART3 and SART3·U4/U6 snRNP complexes accumulate [Stanek, 2017 #4139] (Line 409)

45. 11) Line 430 the authors state “This effect may be exacerbated in tissues such as the testis and CNS with high proliferation rates or which rely heavily on splicing”. Testis and CNS are not organs with high proliferation rates. The CNS is made of mostly post-mitotic neurons; in the testis, the only actively proliferating cells are the spermatogonia. Thus, the statement is incorrect.

Thank you, you are right. We were referring in this case to these tissues during development, but have now removed this statement.

Regarding the dependency on splicing of these two organs, the authors should refer to specific studies that indicated it.

We have now included several references in the text

LINE 501 Indeed, the brain and testis largely exceed the other tissues with respect of splice-variants expression⁵⁶⁻⁵⁹, and higher expression of specific signatures of splice factors suggests that alternative splicing plays a particularly crucial role in these tissues⁶⁰.

56. Soumillon, M., *et al.* Cellular source and mechanisms of high transcriptome complexity in the mammalian testis. *Cell Rep* **3**, 2179-2190 (2013).
57. Merkin, J., Russell, C., Chen, P. & Burge, C.B. Evolutionary dynamics of gene and isoform regulation in Mammalian tissues. *Science* **338**, 1593-1599 (2012).
58. Barbosa-Morais, N.L., *et al.* The evolutionary landscape of alternative splicing in vertebrate species. *Science* **338**, 1587-1593 (2012).
59. Yeo, G., Holste, D., Kreiman, G. & Burge, C.B. Variation in alternative splicing across human tissues. *Genome Biol* **5**, R74 (2004).
60. Grosso, A.R., *et al.* Tissue-specific splicing factor gene expression signatures. *Nucleic Acids Res* **36**, 4823-4832 (2008).

Reviewer #3 (Remarks to the Author):

In this manuscript, Ayers et al reported bi-allelic variants in SART3 gene from six families with children demonstrating 46,XY gonadal dysgenesis, developmental delay and intellectual disability. Further, all patient variants affect highly conserved amino acids in protein domains essential for SART3 function. Using six families and CRISPR/Cas9-engineered iPSC, they further examined and found that one disease-

causing mutant (p.Arg836Gln) exhibit disrupted differentiation into testis and neural cell lineages. By using RNA sequencing and LC-MS/MS, in the differentially expressed gene, cancer signaling and stem cell pluripotency gene are significantly enriched as well as the spliceosome genes. And a highly downregulated genes NOVA2 which may lead to neurodevelopmental disease through interacting with SART3. Nevertheless, SART3 protein is not mis-localized in p.Arg836Gln iPSCs. And the introduction of SART3 patient variants in mouse models results in embryonic lethality. Consistent with the results in the mutated iPSCs, knock down of Rnp4f, the SART3 orthologue of fly, also resulted in the infertile of male flies and fail of normal neural development during embryogenesis.

Overall, this is a well-organized manuscript with significant study that identifies many pathogenic variants in SART3, and their effects to gonadal and neural development. This work is also important for understanding spliceosomopathy caused by SART3 in 46,XY DSD patient. However, several concerns should be addressed.

Major concerns:

46. 1. Among those bi-allelic variants in SART3 gene, why did the authors choose p.Arg836Gln as the study target, why not other variants? For example, the variant in the TUN1.1 family.

We thank the reviewer for this question. The P.Arg836Gln variant was prioritised as it was the first variant we identified, it had the most convincing genetic data for causation (found in two families, perfect segregation of variant with affected/unaffected siblings), and it affected a well-established domain essential for SART3 function. We have now clarified this in the text, Line303 “This variant was prioritised as it had robust genetic data for causation from two families and affects a well-established domain essential for SART3 function.”

47. 2. Instead of using the KD Drosophila strain for investigation, why didn't make a point mutant strain according to the SART3 mutations in human patients?

We agree that a patient variant disease model in Drosophila would have been preferable to RNAi based experiments, however, whilst the Drosophila SART3 ortholog RNP4f is highly conserved in its function as a spliceosome factor and has conservation of several of the protein domains, its conservation at the protein level is low (with an identity of 20%, and similarity of 34%). Unfortunately, none of the amino acids affected in our patients were conserved, meaning we could not create a knock-in fly. Nevertheless, our observation in iPSCs that at least the p.Arg836Gln variant may lead to a reduction in SART3 mRNA and protein means that a KD approach could be an appropriate disease model for some of the variants.

We have now clarified these limitations and why we chose an RNAi approach in the text Line 242 “as poor protein conservation between human SART3 and Rnp4f (20% identity, 34% similarity) precluded us from introducing patient variants. “

48. 3. Since SART3 has two RRM domains for interaction with U6 snRNA, and the Arg836Gln mutation caused significantly changed expression of spliceosome genes in the iPSCs, it is necessary to analyze their RNA-seq data to see how much splicing events were differentially changed.

We agree. New analysis of our RNA-sequencing data using rMATS found little indication that there was a disruption to splicing (i.e. there was no significant differences in the number alternative splicing events between the unedited control and variant iPSCs, specifically exon skipping, alternative 5 or 3' splice sites, mutually exclusive exons or retained introns were not more or less common). We have mentioned this finding in the text as data not shown. We have changed our discussions to include this, pointing out that whilst we have evidence for large scale changes in gene expression and differential transcript usage, we did not detect significantly more splicing events in the variant cell lines. We speculate this may be due to the upregulation of spliceosome factors which may compensate for SART3 reduction in function, as has been observed/proposed in the zebrafish previously. We have also discussed how it may be possible that it is a function of cell type, and that if we had access to patient lines or tissues affected in the patients splicing disruption may be observed in these.

This has been indicated in the text LINE 379-385 rMATS analysis [Shen, 2014 #4142] did not find significant differences in the number of splicing events such as exon skipping or intron retention between control and SART3 homozygous variant iPSCs (data not shown), but differential transcript usage (DTU) analysis using equivalence classes³⁴ found 347 transcripts from 232 genes with DTU between control and homozygous cells (FDR <0.05) (Supp Table 5). Thus changes in gene expression and differential transcript usage are associated with the homozygous SART3 variant in iPSCs.

Minor:

49. 1. Size of the Table 1 should be

We have tried to change the table size to be smaller whilst including additional information as requested by other reviewers.

50. 2. For many figures, labelling of panel letters and scale bars are not in consistent ways.

Thank you we have corrected this now

51. 3. For scales, the "um" should be "µm".

We have corrected this now

Reviewer #4 (Remarks to the Author):

RNA Sequencing was carried out at the Victorian Clinical Genetics Services (VCGS), with Illumina stranded mRNA library prep and sequencing of paired 150 base-pair paired end reads and 30 million reads per sample on the Illumina Novaseq6000.

Fastq files are provided for each sample.

In their manuscript "Variants in SART3 cause a novel spliceosomopathy characterized by failure of testis development and neuronal defects" by Ayers and colleagues focusses on the role on SART3 variants in failures related to testicular development and neuronal defects. The authors identified recessive variants in the SART3 in 9 patients with 46,XY gonadal dysgenesis, intellectual disability, developmental delay and brain anomalies. The conserved role of SART3 was studied in knockdown of the Drosophila orthologue of SART3, due to the fact that null and knock-in patient alleles were lethal in mice. Furthermore, in vitro differentiation of hiPSCs into gonadal and neuronal cells was used to study the role of SART3 variants on testicular and neuronal development.

General comments:

Overall, the study is well-written and deals with an interesting aspect related to failure of the male reproductive organ and neuronal defects. The authors employed several experimental strategies to elucidate the role of SART3 variants, which are difficult to study, especially in humans. However, to improve the content of the current work and to increase the benefit for the reader, I suggest to consider my comments listed below.

Specific comments:

Results:

52. In figure 1A; please add the karyotype of the male offspring of the "ISR3-couple", or add the information, why the karyotyping has not been performed.

Unfortunately genotyping for the unaffected siblings in both ISR3 and FRA1 are not available due to parental wishes. We have now indicated this in the pedigree and in the text "Genetic testing is unavailable for unaffected siblings in FRA1 and ISR3 (Figure 1a)."

53. The histological pictures shown in figure 1b-g are not suitable for publication, due to low quality. Higher magnification as well as improved picture quality is needed to appreciate the results described in the text. In addition, to higher magnifications, overview pictures with the same magnification need to be added.

Unfortunately as this patient was seen many years ago, these are the only images we have for histology. We have included them at higher resolution again and hope the picture quality is improved now. We have also corrected scale bars where possible and have included arrows and inserts/lines to indicate the regions of interest or where the higher magnification images fall on the overview pictures.

54. Page 8, Line 162: Please specify "evidence of some early testicular function" more in detail. What does "some" include?

Thank you. We have now clarified this to read “The other 46, XY children have gonadal dysgenesis but with evidence of some testicular function/testosterone production during development indicated by ambiguous genitalia or virilisation and a lack of Müllerian structures”

55. Page 8, Line 168: The mentioned “small areas of seminiferous tubules” are not clearly shown in figure 1b-g (see also comment above). Please revise figure 1 and improve the picture quality.

Thank you, yes indeed this was our mistake and seminiferous tubules are not present in these images. We have corrected this in the text.

56. Page 8, Line 171: Please add reference values for the gonadotropins measured, to allow the reader to appreciate, what is meant by “low (levels of) gonadotropins”. In this respect, I strongly suggest to add data from healthy controls or reference values (e.g., hormone reference data, frequency of DSDs mentioned) to allow the reader to appreciate the full data set.

We agree with the reviewer and had only provided reference data for hormones in the Supp Table 1 due to size constraints for Table 1. However, we have now moved this information to the main Table 1 where possible. The frequency of 46,XY gonadal dysgenesis has now been provided in the introduction.

57. Page 11, Line 249: Please describe the effect on KD on the somatic cell compartment mentioned, more in detail. Which somatic cells are affected? Is there a direct effect on the germ cells or via the somatic environment building the stem cell niche? Is there any connection to the Sox9 (a Sertoli cell marker also mentioned in the introduction of the manuscript in another context)?

We thank the reviewer for these questions. We have now elaborated on the somatic cell compartment KD, detailing the exact expression of the Gal4 driver used in these experiments and the phenotype.

LINE 257-269. “Testicular somatic cells offer essential support to the germline during spermatogenesis in flies and humans. Our 46,XY patient phenotypes suggest a role for SART3 in the somatic cells of the testis, as a germline only disruption would not result in gonadal dysgenesis. To assess whether Rnp4f is functioning in the somatic cells, RNAi transgenes were expressed using the somatic cell-specific driver traffic jam-Gal4 (tj-Gal4). This resulted in large, immature, misshaped testes (Figure 3l). No disruption to the somatic cells were observed (data not shown), but changes in spermatogenesis were similar to those found in the global KD flies. Phase contrast imaging suggested an accumulation of early germ cells/spermatocytes and very few motile sperm in the somatic KD testis (Figure 3k), compared to controls (Figure 3i). Immunofluorescent using the Don Juan-GFP, a marker of elongated spermatids, found a loss of these in the KD (Figure 3j,l). Thus, SART3 appears to have a conserved role in Drosophila testis development, where it is required in the somatic gonadal cells for proper spermatogenesis. “

SOX9 does not play a conserved role in Drosophila, and therefore all we can derive from this data is that SART3 is similarly required for proper testis development in flies and humans.

58. Page 12, Line 276 & 279: Please add the information regarding the species, here.

Due to potential differences in line number we weren't entirely clear about where the reviewer is referring to; was it the species of the iPSC line or the passage above this - "early lethality associated with loss of SART3"? Nevertheless we have clarified species in both places now.

59. Page 13, Line 282 – 283: Please add the details regarding the markers and expression profile (e.g., RNA, DNA or protein level) as supplementary data.

Apologies, these were previously in the Supp. data but now we have included a reference to them in the text (Supp. Figure 4c-f) and elaborated on their use in the figure legend. We have added antibody details for the FACS analysis to the supplementary antibodies table.

60. Figure 4f-n: Please add higher magnifications here, as well.

We have reworked Figure 4/gonad organoid results – and have included additional images with higher magnification.

61. Page 13, Line 290-291: Please add the information regarding the functionality tests used for these organoids. As it appears from the text and figures, the similarities to the organoids described in Knarston et al. 2020 are rather limited.

62. Page 13, Line 303-304: The results described are not enough to confirm the statement made, since all markers are also expressed in KD, and structural features can vary under in vitro conditions. In order to confirm the results, the functionality of the organoids need to be demonstrated (see also comment above).

We thank the reviewer for these comments. As we detail in our Stem Cell Reports paper, our human gonadal organoid protocol does not recapitulate a fully functional testis, and our "functionality tests" are limited to gene and protein expression and organoid size and structure in this model. While we realise this is a limitation, in the absence of human fetal testis cell lines or other organoid models for the human fetal gonad, we believe our organoid model is a valuable tool for working towards an understanding of how SART3 variants might disrupt testis development.

We have now reworked the testis organoid section to include additional analyses including more imaging, size analysis from a greater number of organoids and qPCR during the differentiation process in the three lines.

*Given the limitations of our model we have included discussion of this. LINE 476
“Optimisation of our basic testis organoid model to include additional cell types and
functional readouts may provide better insight into disease mechanisms in the future.
“*

63. Suppl. Figure 4d shows morphological impairments/differences. Therefore, tests of used iPSCs, incl. teratoma-formation assay, in addition, to details of marker panels used for the different cells need to be added. In this respect, the effect of SART3 variants on the undifferentiated iPSCs needs to be evaluated. How much do cell line specific features affect the ability of the cells to differentiated into gonadal and neuronal cells?

We were interested that the reviewer has suggested morphological impairments in the iPSCs, as we had not observed this ourselves. Thus we have included better images of the iPSCs and OCT3/4 staining now in Figure 4d as we realised that the several images were missing scale bars and the first immunofluorescence image had been somewhat corrupted.

We agree that this assessment is essential. We have now added additional information about the FACS analysis for pluripotency (Supp Figure 4 legend, materials and methods, antibody table)

We have also included the results of embryoid body creation from each iPSC line. These show the SART3 heterozygous and homozygous variant iPSC lines efficiently differentiated into the three germ layers in a manner comparable to the unedited control line (Supp. Figure 4q-y). Thus we do not believe the iPSC lines to have gross impairment in their ability to differentiate, although RNA-seq and mass spec data has indicated significant changes to signalling in these undifferentiated cells, and some interesting differences such as an increase in OCT4 expression, which may impede specific developmental trajectories.

64. Page 14, Line 323-324 and related to suppl figure 4 & 5; fig.4: What is the effect of SART3.p.R836Q Het in terms of gonadal cell differentiation?

As mentioned above the reworked testis organoid section now includes more data relating the heterozygous line in both Figure 4 and Figure 5. The heterozygous line, as with the neuronal differentiation, appears to differentiate well and does not appear to undergo the significant apoptosis seen in the homozygous. Although it does demonstrate some differences in the expression of lineage or testis genes in line with the homozygous line.

65. Page 17: The authors mention that NT2/D1 is cell line used as model for somatic Sertoli cells. However, this cell line is a multipotent cell line derived from human embryonal carcinoma cells (a rather uncommon germ cell tumor type) and therefore not really optimal to model somatic cells, such as Sertoli cells. Therefore, additional references should be included, showing the suitability of the cell line for the performed studies.

Thank you. Indeed the NT2/D1 cell line has some limitations as a model for testis development, however in the absence of other more representative human fetal testis cell culture lines this has been used, and is accepted, in the field as a model to interrogate the signalling of testis pathway genes. We have included the following references to illustrate this; Knowler et al 2007, a study that looked at gene expression in this line and concluded that “Cell-specific markers demonstrate that NT2/D1 cells reflect a number of cell types in the gonad including Sertoli, Leydig and germ cells..and that male pathways initiated by SRY, SOX9 and SF-1 remain intact in these cells.”

Ludbrook et al 2016 where the authors looked at transcriptional targets of the SOX9 gene, and a more recent publication by Stewart et al 2021 that used this cell line to investigate how Oestrogen regulates SOX9 bioavailability and ERK signalling.

The text now reads LINES 411-413 NT2/D1 cells have also been employed as a model for testis-somatic cells as they express at least 40 testis-specific genes and male pathways initiated by SRY, SOX9 and SF-1 are intact ³⁸[Ludbrook, 2016 #253][Stewart, 2021 #4140].”

66. Page 17, Line 382-383: No effects on genes important for testicular development, such as NR5A1, WT1 and SOX9 were observed, which is very interesting as those include the main players, such as SOX9 for early male gonadal formation. Please discuss this as well as the effect on FGF9 in relation to the before mentioned genes more in detail in the discussion.

As you will see in our reworked organoid section, we have found evidence in this model for changes in additional genes important in development of the bi-potential gonad and testis, including NR5A1, GATA4 and FGF9. The NT2/D1 study found similar results in GATA4 and FGF9, but as the reviewer mentioned, did not show changes in other genes such as SOX9. This is interesting, and so as the reviewer mentioned we have now included more of a discussion on this. Lines 464-478

Material and methods:

Please add the information regarding all negative and positive controls used (incl. samples, antibodies, concentrations) for the different staining performed in the study used to validate the antibodies used.

All antibody details have been provided as part of the supplementary data (Table 7) including source and concentration used. Information regarding each antibody used and validation is also provided in the Nature Communications Reporting Summary. All antibodies used are commercial and have been previously validated and published. We have now also included details of controls where applicable in materials and methods.

REVIEWER COMMENTS

Reviewer #1 (Remarks to the Author):

The authors have been very responsive to the reviewers' critiques and the quality of the manuscript has been improved significantly.

Despite the editing, there remain significant errors in capitalization, consistency of tense usage, singular-plural agreement, and punctuation. The authors are urged to consult with a local editor to ameliorate these grammatical and usage issues and, thus, make the manuscript suitable for publication.

Reviewer #2 (Remarks to the Author):

The authors have addressed most of the concerns raised to the original manuscript and this revised version is greatly improved.

Reviewer #3 (Remarks to the Author):

I think that the revisions that authors have made and incorporated into this version of the manuscript have significantly improved. No further questions.

Reviewer #4 (Remarks to the Author):

The authors addressed all my comments. No further questions from my side.

Reviewer #5 (Remarks to the Author):

The manuscript by Ayers et al (Variants in SART3 cause a novel spliceosomopathy characterised by failure of testis development and neuronal defects) identifies variants of SART3 gene in human patients. The authors developed genetic and in vitro models (of the pathogenic variants) to perform a series of systematic experiments documenting the relevance of the variants to disease pathology.

The experiments are logical, and they establish the observations and interpretations convincingly.

While the other reviewers (with extensive expertise in the subject area) seem to have provided several valuable critiques, related to overall interpretations and conclusions of the study, this reviewer evaluated the gene editing experiments used for generating animal models and cell lines (which serve as critical reagents for the study).

Below are some comments and suggestions that may help improve the manuscript further (and/or perhaps help authors to plan their future experiments).

Regarding the (failed) mouse model generation experiments, the information provided in supplementary table 4 shows that at least two different transgenic core facilities attempted generating at least two different knock-in models (injecting as many as 3000 zygotes and screening about 50 live born offspring). The reviewer would like to provide some thoughts as to why the experiments were unsuccessful and make suggestions that may help authors generate the models if they plan to do those experiments in the future. The reviewer does not think that the authors should generate the models by modifying their gene editing experiments (for this paper) because the authors have generated hiPSCs to accomplish the same goal.

A couple observations (about the failed attempts) do not make full sense. For example: (a) if a similar strategy (of creating KI mutations) in hiPSCs was successful, it could have been possible to create mouse models too, but it is understandable that some genomic loci are highly challenging to create knock-in models. (b) The overall birth rate (50/3000, which is <2%) is significantly lower because typical birth rate in gene editing experiments fall in the range of 10 to 30%.

One of the major factors that determines the success of creating KI models is the distance between the guide cleavage site and the site of new change (to be inserted in the genome). If the distance is a few bases (like >4 or 5 nucleotides), the KI efficiency drops significantly. Based on the information available in the supplementary table 4 and the materials and methods in the supplementary files, the guides they chose (for one of the knock-in experiments) indeed cleave very far away from the insertion site (29 bases or 19 bases). Another factor that contributes to lower efficiency, generally, is the guide cleavage efficiency. based on the (lower) number of indel containing mice, the efficiency of the guides may also have been poor. The authors seem to have desperately tried many things (including truncated guides and Cas9 nickase in some cases), but it is unfortunate that the experiments did not work.

If the authors plan to extend this study for future projects, here are some suggestions: (1) they may be able to overcome the (mouse model generation) challenge by choosing two guides (flanking the insertion site) about 40 or 50 or 100 nucleotides apart from each other and remove the genomic segments between them and insert a donor containing the desired nucleotide change at the target site. (2) If they want to ensure the guide cleavage efficiencies (of the new guides) before proceeding to generating the models it may help to find the best guides for the purpose. (3) The authors can use newer versions of CRISPR reagents like RiboNuceloProteins (RNPs). The authors have used 'CRISPR reagent formats' that are not popular anymore. Specifically, they used in vitro transcription generated guide RNAs and Cas9 mRNAs, which were commonly used until about 5 years ago (before RNPs became popular). Some batches of in-house generated RNAs (unless purified extensively) can contain some impurities that may affect zygote viability. It is likely that their very low birth rate (<2%) may be attributable to the toxicity of such reagents(?).

Considering all these points, the reviewer strongly suggests that the statement in lines 295-297 should be deleted (...We concluded it likely that these patient SART3 variants are incompatible with viability in mice, and we. therefore sought additional human cell-based models in which to investigate patient variants....).

Summary of the comment (about the mouse model generation experiments):

- i) it is intriguing that multiple attempts to generate knock-in mouse models were unsuccessful but it is not necessary (for this manuscript) to generate such models
- ii) some modifications in model generation experiments could help the authors if they choose to pursue such projects in the future.

hiPSC cell line generation experiments seem to have worked as expected. Two minor suggestions:

- a. Line 875, where the guide RNA sequence is mentioned, the authors can consider indicating the distance (in nucleotides) this guide cleaves the genomic site from the insertion site.
- b. Line 886, it would be better to describe some more details about isolating single cell clones such as, did the authors pick the colonies into 96 well plates and then expanded them to 24 well plates and how many colonies were screened before they got heterozygous and homozygous cell lines.

For the readers, it will certainly look surprising that authors were able to get the knock-in cell lines because creating cell lines is much harder than generating mouse models via pronuclear injection. The readers would wonder how did authors were successful in generating both mono-allelic and bi-allelic

modifications in cells when they were not successful in generating even mono-allelic insertion in mouse zygotes.

It is possible that authors were successful because the guide RNAs for the human genomic sites may be more efficient and they are much closer to the modification sites or simply that the approach (of plasmid mediated delivery of CRISPR tools) may have been less toxic (compared to high level toxicity of in house generated RNAs used for mouse zygote injections).

.

Based on all these thoughts (about both mouse and human cell line gene editing experiments) the reviewer suggests authors to consider one of the below two alternative options:

A. To discuss the reasons why their mouse model generation experiments were unsuccessful and why their hiPSc model generation experiments worked (perhaps by discussing the points mentioned above in this review comments)

Or

B. To completely remove the mouse model generation experiments and only mention hiPSC experiments. The advantage of option B is that it will help shorten the article by cutting down the experiments that did not work.

If authors prefer to retain the mouse model generation experiments, this reviewer is of strong opinion that the due credit should be given to the technicians who performed the (painstaking mouse) experiments by adding them as authors (if they are not already included) but the reviewer will leave it for authors' discretion.

All the best,

CB Gurumurthy.

REVIEWER COMMENTS and responses

Please note – lines refer to marked up version of word document

Reviewer #1 (Remarks to the Author):

The authors have been very responsive to the reviewers' critiques and the quality of the manuscript has been improved significantly.

Despite the editing, there remain significant errors in capitalization, consistency of tense usage, singular-plural agreement, and punctuation. The authors are urged to consult with a local editor to ameliorate these grammatical and usage issues and, thus, make the manuscript suitable for publication.

Thank you, we have done as suggested and addressed grammatical and usage issues.

Reviewer #2 (Remarks to the Author):

The authors have addressed most of the concerns raised to the original manuscript and this revised version is greatly improved.

Thank you

Reviewer #3 (Remarks to the Author):

I think that the revisions that authors have made and incorporated into this version of the manuscript have significantly improved. No further questions.

Thank you

Reviewer #4 (Remarks to the Author):

The authors addressed all my comments. No further questions from my side.

Thank you

Reviewer #5 (Remarks to the Author):

The manuscript by Ayers et al (Variants in SART3 cause a novel spliceosomopathy characterised by failure of testis development and neuronal defects) identifies variants of SART3 gene in human patients. The authors developed genetic and in vitro models (of the pathogenic variants) to perform a series of systematic experiments documenting the relevance of the variants to disease pathology.

The experiments are logical, and they establish the observations and interpretations convincingly.

While the other reviewers (with extensive expertise in the subject area) seem to have provided several valuable critiques, related to overall interpretations and conclusions of the study, this reviewer evaluated the gene editing experiments used for generating animal models and cell lines (which serve as critical reagents for the study).

Below are some comments and suggestions that may help improve the manuscript further (and/or perhaps help authors to plan their future experiments).

Regarding the (failed) mouse model generation experiments, the information provided in supplementary table 4 shows that at least two different transgenic core facilities attempted generating at least two different knock-in models (injecting as many as 3000 zygotes and screening about 50 live born offspring). The reviewer would like to provide some thoughts as to why the experiments were unsuccessful and make suggestions that may help authors generate the models if they plan to do those experiments in the future. The reviewer does not think that the authors should generate the models by modifying their gene editing experiments (for this paper) because the authors have generated hiPSCs to accomplish the same goal.

A couple observations (about the failed attempts) do not make full sense. For example: (a) if a similar strategy (of creating KI mutations) in hiPSCs was successful, it could have been possible to create mouse models too, but it is understandable that some genomic loci are highly challenging to create knock-in models. (b) The overall birth rate (50/3000, which is <2%) is significantly lower because typical birth rate in gene editing experiments fall in the range of 10 to 30%.

One of the major factors that determines the success of creating KI models is the distance between the guide cleavage site and the site of new change (to be inserted in the genome). If the distance is a few bases (like >4 or 5 nucleotides), the KI efficiency drops significantly. Based on the information available in the supplementary table 4 and the materials and methods in the supplementary files, the guides they chose (for one of the knock-in experiments) indeed cleave very far away from the insertion site (29 bases or 19 bases). Another factor that contributes to lower efficiency, generally, is the guide cleavage efficiency. based on the (lower) number of indel containing mice, the efficiency of the guides may also have been poor. The authors seem to have desperately tried many things (including truncated guides and Cas9 nickase in some cases), but it is unfortunate that the experiments did not work.

If the authors plan to extend this study for future projects, here are some suggestions: (1) they may be able to overcome the (mouse model generation) challenge by choosing two guides (flanking the insertion site) about 40 or 50 or 100 nucleotides apart from each other and remove the genomic segments between them and insert a donor containing the desired nucleotide change at the target site. (2) If they want to ensure the guide cleavage efficiencies (of the new guides) before proceeding to generating the models it may help to find the best guides for the purpose. (3) The authors can use newer versions of CRISPR reagents like RiboNuceloProteins (RNPs). The authors have used 'CRISPR reagent formats' that are not popular anymore. Specifically, they used in vitro transcription generated guide RNAs and Cas9 mRNAs, which were commonly used until about 5 years ago (before RNPs became popular). Some batches of in-house generated RNAs (unless purified extensively) can contain some impurities that may affect zygote viability. It is likely that their very low birth rate

(<2%) may be attributable to the toxicity of such reagents(?).

Considering all these points, the reviewer strongly suggests that the statement in lines 295-297 should be deleted (...We concluded it likely that these patient SART3 variants are incompatible with viability in mice, and we. therefore sought additional human cell-based models in which to investigate patient variants...).

We thank this reviewer very much for these highly insightful comments.

- *We have removed lines 295-297 as suggested, as we agree there may have been other factors contributing to our inability to create a mouse model.*
- *We have now changed the title of this section from “Introduction of SART3 patient variants in mouse models results in embryonic lethality” to “Genome editing failed to introduce patient SART3 variants in mice LINE 384*

Summary of the comment (about the mouse model generation experiments):

- i) it is intriguing that multiple attempts to generate knock-in mouse models were unsuccessful but it is not necessary (for this manuscript) to generate such models
- ii) some modifications in model generation experiments could help the authors if they choose to pursue such projects in the future.

hiPSC cell line generation experiments seem to have worked as expected. Two minor suggestions:

a. Line 875, where the guide RNA sequence is mentioned, the authors can consider indicating the distance (in nucleotides) this guide cleaves the genomic site from the insertion site.

The cut site is 1 bp away from the intended c.2507G>A conversion. We have now included this in the materials and methods. LINE 1240-1241

b. Line 886, it would be better to describe some more details about isolating single cell clones such as, did the authors pick the colonies into 96 well plates and then expanded them to 24 well plates and how many colonies were screened before they got heterozygous and homozygous cell lines.

- *For the specific details pertaining to isolating / screening clones etc we have cited the Nature Protocols paper authored by Sara Howden, who runs the gene editing facility at MCRI and created the lines (Howden et al, 2018). Using this method (simultaneous reprogramming and gene-editing) all iPSC colonies that are picked are clonal and therefore don't require single cell cloning. They typically pick 48 colonies and handover all positives. In 48, one homozygous clone and two heterozygous clones were found in our case.*
- *We have included the following “Using this method (simultaneous reprogramming and gene-editing) all iPSC colonies picked are clonal and therefore don't require single cell cloning. 48 colonies were picked and screened.” LINES 1252-1254*

For the readers, it will certainly look surprising that authors were able to get the knock-in cell lines because creating cell lines is much harder than generating mouse models via pronuclear injection. The readers would wonder how did authors were successful in generating both

mono-allelic and bi-allelic modifications in cells when they were not successful in generating even mono-allelic insertion in mouse zygotes.

It is possible that authors were successful because the guide RNAs for the human genomic sites may be more efficient and they are much closer to the modification sites or simply that the approach (of plasmid mediated delivery of CRISPR tools) may have been less toxic (compared to high level toxicity of in house generated RNAs used for mouse zygote injections).

Based on all these thoughts (about both mouse and human cell line gene editing experiments) the reviewer suggests authors to consider one of the below two alternative options:

A. To discuss the reasons why their mouse model generation experiments were unsuccessful and why their hIPSc model generation experiments worked (perhaps by discussing the points mentioned above in this review comments)

Or

B. To completely remove the mouse model generation experiments and only mention hIPSC experiments. The advantage of option B is that it will help shorten the article by cutting down the experiments that did not work.

If authors prefer to retain the mouse model generation experiments, this reviewer is of strong opinion that the due credit should be given to the technicians who performed the (painstaking mouse) experiments by adding them as authors (if they are not already included) but the reviewer will leave it for authors' discretion.

- *Thank you for these excellent suggestions to improve the manuscript. We have chosen option A - to leave in the mouse work in the manuscript as we believe that it is important to show that we attempted a mouse model and indicate that there may be potential difficulties in mouse generation for any researchers who would like to attempt a SART3 mouse model in the future. We now also suggest how these challenges could be ameliorated by using different strategies as suggested by the reviewer. In addition, our inability to generate a mouse model was one of the reasons we employed the human iPSC system.*
- *It is also interesting to note, although we do not believe it deserves inclusion in the paper, that in 2014 we successfully introduced a different variant in SART3 exon 1 into the mouse. This was a heterozygous variant in a patient which was ultimately shown to be a non-pathogenic polymorphism. Compared to the other patient variants, this worked very easily, and we got homozygous mice very quickly, although they had no significant phenotype – consistent with it being benign in the patient. This was another reason that we had the impression that other variants were deleterious to survival in mouse, but we agree with the suggestions given on how best to improve our strategy. Indeed, the CRISPR mouse work was attempted in 2014 (Univ. QLD), 2015 (Walter & Eliza Hall Inst.) and 2017 (Univ.QLD). This was when in vivo Crispr was relatively new, and therefore the use of more refined modern approaches might prove successful if attempted today.*

➤ *The discussion now reads...(LINES 582-588)*

“Our Drosophila studies demonstrated a critical and conserved requirement for SART3 in testicular and neuronal development. We were unable to further study this condition in mice as, despite multiple attempts, our CRISPR/Cas9 genome editing experiments failed to produce viable offspring carrying either the p.Arg836Gln or p.Glu211Lys patient variant. This is somewhat surprising given that we successfully introduced the p.Arg836Gln change in iPSCs but may be due to differences in guide RNA efficiency or some other factor. Indeed, mouse experiments were carried out when in vivo CRISPR/Cas9 was a relatively new tool, and using more recently refined approaches may yield better success.”

REVIEWERS' COMMENTS

Reviewer #5 (Remarks to the Author):

Glad to hear that the comments were helpful to improve the manuscript.

The manuscript is further improved in this submission.

Two (very) minor suggestions:

1. Line 385: Instead of the heading "Genome editing failed to introduce patient SART3 variants in mice" the authors may consider something like "Attempts to generate SART3 variants knockin mouse models were unsuccessful". This would mean that 'attempts' failed, not genome editing failed!

2. Line 587: The sentence "Indeed, mouse experiments were carried out when in vivo CRISPR/Cas9 was a relatively new tool, and using more recently refined approaches may yield better success." can be revised (with some references added) as follows:

Of note, we attempted CRISPR/Cas9 mouse model generation experiments when the tool was relatively new. The guide RNA and Cas9 mRNAs were generated using in vitro transcription method (you may cite one or two references about earlier protocols). The overall birth rates of our CRISPR experiments were low (~2%), which suggest that our batches of in vitro transcription reagents may have had impurities causing embryo toxicity. In the recent few years, ribonucleoprotein forms of CRISPR are shown to produce consistent results with higher editing efficiencies (you may cite one or two references about RNP based mouse model generation protocols). It is likely that refined CRISPR reagent formats, such as ribonucleoproteins, may yield better success in generating the models"

Reviewer #5 (Remarks to the Author):

Glad to hear that the comments were helpful to improve the manuscript.

The manuscript is further improved in this submission.

Two (very) minor suggestions:

1. Line 385: Instead of the heading “Genome editing failed to introduce patient SART3 variants in mice” the authors may consider something like “Attempts to generate SART3 variants knockin mouse models were unsuccessful”. This would mean that ‘attempts’ failed, not genome editing failed!

Thank you we made this change.

2. Line 587: The sentence “Indeed, mouse experiments were carried out when in vivo CRISPR/Cas9 was a relatively new tool, and using more recently refined approaches may yield better success.” can be revised (with some references added) as follows:

Of note, we attempted CRISPR/Cas9 mouse model generation experiments when the tool was relatively new. The guide RNA and Cas9 mRNAs were generated using in vitro transcription method (you may cite one or two references about earlier protocols). The overall birth rates of our CRISPR experiments were low (~2%), which suggest that our batches of in vitro transcription reagents may have had impurities causing embryo toxicity. In the recent few years, ribonucleoprotein forms of CRISPR are shown to produce consistent results with higher editing efficiencies (you may cite one or two references about RNP based mouse model generation protocols). It is likely that refined CRISPR reagent formats, such as ribonucleoproteins, may yield better success in generating the models”

We thank the reviewer for this. We have revised the text and included new references (Yang et al 2014, Quadros et al), – given text limits we have now written:

“Indeed, we attempted CRISPR/Cas9 mouse model generation when the tool was relatively new and guide RNA and Cas9 mRNAs were generated using an *in vitro* transcription method [Yang, 2014 #4199], which can introduce impurities causing embryo toxicity. More recently refined approaches such as ribonucleoprotein forms of CRISPR, which have been found to produce consistent results with higher editing efficiencies (Quadros et al), may yield better success.”